# PERTURB-AND-COMPARE APPROACH FOR DETECTING OUT-OF-DISTRIBUTION SAMPLES IN CONSTRAINED ACCESS ENVIRONMENTS

## ABSTRACT

Accessing machine learning models through remote APIs has been gaining more prevalence following the recent trend of scaling up model parameters for increased performance. Even though these models exhibit remarkable ability, detecting out-of-distribution (OOD) samples is still an important issue concerning the safety of the end users, as these samples may induce unreliable outputs from the model. In this work, we propose an OOD detection framework, MixDiff, that is applicable even when the model parameters or its activations are not accessible to the end user. To bypass the access restriction, MixDiff applies an identical input-level perturbation to a given target sample and an in-distribution (ID) sample that is similar to the target and compares the relative difference of the model outputs of these two samples. MixDiff is model-agnostic and compatible with existing output-based OOD detection methods. We provide theoretical analysis to illustrate MixDiff's effectiveness at discerning OOD samples that induce overconfident outputs from the model and empirically show that MixDiff consistently improves the OOD detection performance on various datasets in vision and text domains.

## 1 INTRODUCTION

Recent developments in deep neural networks (DNNs) opened the floodgates for a wide adaptation of machine learning methods in various domains such as computer vision, natural language processing, and speech recognition. As these models garner more users and widen their application area, the magnitude of impact that they may bring about when encountered with a failure mode also increases. One of the causes of these failure modes is when an out-of-distribution (OOD) sample is fed to the model. These samples are problematic because DNNs often produce unreliable outputs if there is a large deviation from the in-distribution (ID) samples that the model has been validated to perform well.

OOD detection is the task of determining whether an input sample is from ID or OOD. Several studies explore measuring how uncertain a model is about a target sample relying on the model's output (Hendrycks & Gimpel, 2017; Liu et al., 2023b). While these methods are desirable in that they do not assume access to the information inside the model, they can be further enhanced given access to the model's internal activations, (Sun et al., 2021) or its parameters (Hsu et al., 2020). However, the access to the model's internal states is not always permitted. As the trend of offering machine learning models themselves as a service continues, these models often find themselves interacting with the end user through remote APIs (Sun et al., 2022b). This limits the utilization of rich information inside the model (Huang et al., 2021), as well as the modification possibilities (Sun & Li, 2022) that can be effectively used to detect OOD samples. In this work, we explore ways to bypass this access restriction through the only available modification point, namely, the models' inputs.

Data samples in the real world may contain distracting features that can negatively affect the model's performance. Sometimes these distractors may possess characteristics resembling a class that is different from the sample's true label. In this case, the model's predictions could become uncertain as it struggles to decide which class the sample belongs to. Similarly, the model could put too much importance on a feature that resembles a certain in-distribution characteristic, outputting an

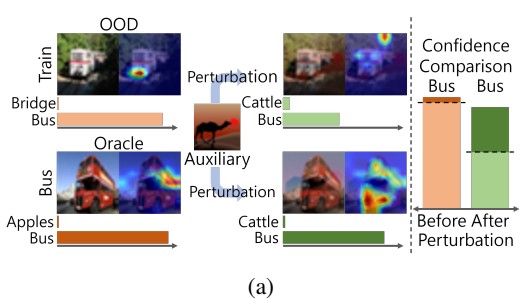 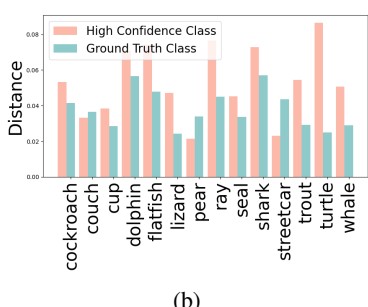

(a)            (b)

Figure 1: **(a):** Class activation map of an OOD sample (train) for the predicted class (bus) exhibits a high degree of sensitivity when an auxiliary image (camel) is mixed to it. The same class activation map of an image of an actual bus is more robust to the same perturbation. (Top 2 classes are shown). **(b):** Average distance of the CAMs of high confidence class and the ground truth class after perturbation (averaged over each OOD class).

overconfident prediction, even though the sample does not belong to any of the classes that the model was tasked to classify.

We start from the intuition that the contributing features in a misclassified sample, either misclassified as ID or OOD, will tend to be more sensitive to perturbations. In other words, these features that the model has overemphasized will be more brittle when compared to the actual characteristics of the class that these features resemble. Take as an example the image that is at the top left corner of Figure 1a. This sample is predicted to be a bus with a high confidence score, despite it belonging to an OOD class train. When we exact a perturbation to this sample by mixing it with some other auxiliary sample, the contribution of the regions that led to the model's initial prediction is significantly reduced as can be seen by the change in the class activation maps (CAM) (Chen et al., 2022b). However, when the same perturbation is applied to an actual image of a bus, the change is significantly less abrupt. The model's prediction scores show a similar behavior. Figure 1b shows that the distance between the CAMs of the unperturbed and perturbed versions of an OOD sample's predicted class tends to be higher compared to its ground truth class, even though the OOD sample had a high confidence score for that class. Experimental details are in Appendix E.8.

Motivated by the above idea, we propose an OOD detection framework, MixDiff, that exploits the perturb-and-compare approach without any additional training. MixDiff employs a widely used data augmentation method Mixup (Zhang et al., 2018) as the perturbation method. Its overall procedure is outlined as follows: (1) perturb a target sample by applying Mixup with an auxiliary sample and get the model's prediction by feeding the perturbed target sample to the model; (2) perturb an ID sample (oracle sample in Figure 1a) of the predicted class by following the same procedure; (3) compare the uncertainty scores of the perturbed samples. By comparing how the model's outputs of the target sample and a similar ID sample behave under the same perturbation, MixDiff augments the limited information contained in the model's prediction scores. This gives MixDiff the ability to better discriminate OOD and ID samples, even when the model's unperturbed prediction scores are almost identical.

We summarize our key contributions and findings as follows: (1) We propose an OOD detection framework, MixDiff, that can enhance existing OOD scores in constrained access environments where only the models' inputs and outputs are accessible. (2) We provide a theoretical insight as to how MixDiff can mitigate the overconfidence issue of existing output-based OOD scoring functions. (3) MixDiff consistently improves various output-based OOD scoring functions when evaluated on OOD detection benchmark datasets in constrained access scenarios where existing methods' applicability is limited.

## 2 RELATED WORK

**Output-based OOD scoring methods**    Various works propose OOD scoring functions measuring a classifier's uncertainty from its prediction scores. Some of these methods rely solely on the model's

prediction probability. Maximum softmax probability (MSP) (Hendrycks & Gimpel, 2017) utilizes the maximum value of the prediction distribution. Steinhardt & Liang (2016); Thulasidasan et al. (2021) use Shannon entropy as a measure of uncertainty, while GEN (Liu et al., 2023b) proposes a generalized version of the entropy score. KL Matching (Hendrycks et al., 2022) finds the minimum KL divergence between the target and ID samples. D2U (Yilmaz & Toraman, 2022) measures the deviation of output distribution from the uniform distribution. If we take a step down to the logit space, maximum logit score (MLS) (Hendrycks et al., 2022) utilizes the maximum value of the logits. Energy score (Liu et al., 2020) takes LogSumExp over the logits for the OOD score. While these output-based methods possess a desirable property, i.e., relaxed assumption on model accessibility, the information contained in them tends to be limited. This motivates us to investigate the perturb-and-compare approach as an additional source of information.

**Enhancing output-based OOD scores** Another line of work focuses on enhancing the aforementioned output-based OOD scores to make them more discriminative. ODIN (Liang et al., 2018) and G-ODIN (Hsu et al., 2020) utilize Softmax temperature scaling and gradient-based input pre-processing to enhance MSP (Hendrycks & Gimpel, 2017). ReAct (Sun et al., 2021) alleviates the overconfidence issue by clipping the model's activations if they are over a certain threshold. BAT (Zhu et al., 2022) uses batch normalization (Ioffe & Szegedy, 2015) statistics for activation clipping. DICE (Sun & Li, 2022) leverages weight sparsification to mitigate the over-parameterization issue. Recently, methods that are based on activation or weight pruning approaches (Djurisic et al., 2023; Ahn et al., 2023) also have been proposed. Our work can be viewed as an OOD score enhancement method in constrained access environments, where models' gradients, activations, and parameters are not accessible, leaving the model inputs as the only available modification point.

**Utilization of deeper access for OOD scoring** Several studies exploit the rich information that the feature space provides when designing OOD scores. Olber et al. (2023); Zhang et al. (2023) utilize ID samples activations for comparison with a target sample. Models' inner representations are employed in methods that rely on class-conditional Mahalanobis distance (Lee et al., 2018; Chen et al., 2022c; Ren et al., 2021). ViM (Wang et al., 2022) proposes an OOD score that complements the Energy score (Liu et al., 2020) with additional information from the feature space. Sun et al. (2022c) use the target sample's feature level KNN distance to ID samples. GradNorm (Huang et al., 2021) employs the gradient of the prediction probabilities' KL divergence to the uniform distribution. Zhang & Xiang (2023) show that decoupling MLS (Hendrycks et al., 2022) can lead to increased detection performance if given access to the model parameters. However, these methods are not applicable to black-box API models where one can only access the model's two endpoints, i.e., the inputs and outputs.

**Robustness to adversarial attacks** Overparameterized DNNs are known to be susceptible to adversarial attacks (Liu et al., 2023a; Wang et al., 2023b). One of such attacks is data poisoning attack where attackers modify parts of the training data with the purpose of inducing certain undesirable outputs from the model at inference time (He et al., 2023). Similar to our approach, data augmentation techniques such as Mixup (Zhang et al., 2018; Yun et al., 2019) are shown to be effective at defending such attacks Borgnia et al. (2021b;a). In OOD detection task, Chen et al. (2022a; 2021); Azizmalayeri et al. (2022) show that OOD detectors can also be susceptible to adversarial attacks, where the target sample is modified for malicious purposes to induce either high or low confidence scores from the classifier model.

## 3 METHODOLOGY

In this section, we describe the working mechanism of MixDiff framework. MixDiff is comprised of the following three procedures: (1) find ID samples that are similar to the target sample and perturb these samples by performing Mixup with an auxiliary sample; (2) perturb the target sample by performing Mixup with the same auxiliary sample; (3) measure the model's uncertainty of the perturbed target sample relative to the perturbed ID samples. We now provide a detailed description of each procedure.

**Oracle-side perturbation** We feed the given target sample, $x_t$, to a classification model $f(\cdot)$ and get its prediction scores for $K$ classes, $O_t$, and the predicted class label, $\widehat{y}_t$, as shown in Equation 1.

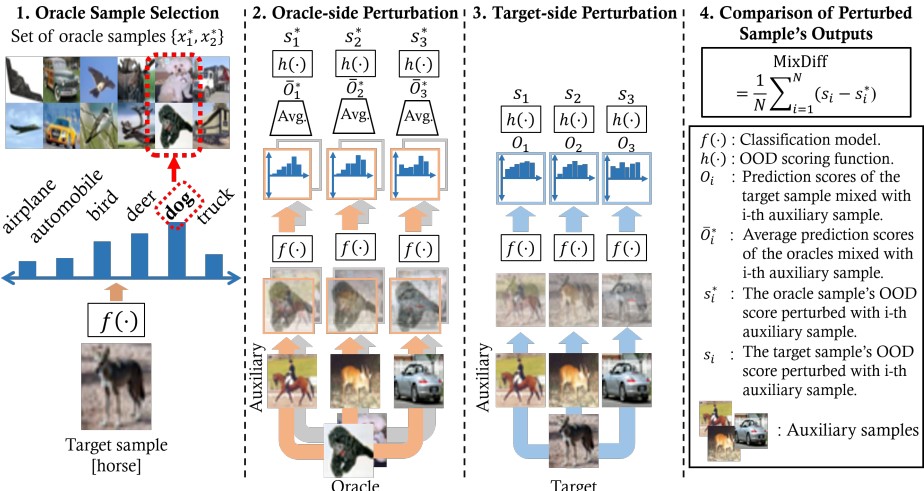

Figure 2: The overall figure of MixDiff with the number of Mixup ratios, $R = 1$, the number of classes, $K = 6$, the number of auxiliary samples, $N = 3$, and the number of oracle instances, $M = 2$. We omit Mixup ratio $r$ for simplicity. **Step 1:** the target sample's prediction is computed from the classifier $f(\cdot)$. The oracle samples corresponding to the predicted class are selected. **Step 2:** the oracle samples are perturbed with the auxiliary samples. Then, we calculate the OOD score for each perturbed oracle sample, followed by computing the average for each auxiliary sample. **Step 3:** we perturb the target sample with the same auxiliary samples. As in the previous step, we compute the OOD score of the perturbed target samples. **Step 4:** we compare the OOD scores of the perturbed target and the perturbed oracles.

$$O_t = f(x_t) \in \mathbb{R}^K, \quad \widehat{y}_t = \arg\max(O_t) \tag{1}$$

Next, we assume a small set of $M$ labeled samples, $\Omega_k = \{(x_m^*, y_k^*)\}_{m=1}^M$, for each class label $k$. We refer to these samples as the oracle samples. From these, we take the samples that are of the same label as the predicted label $\widehat{y}_t$. Then, we perturb each oracle sample, $x_m^*$, by performing Mixup with an auxiliary sample, $x_i \in \{x_i\}_{i=1}^N$, with Mixup rate $\lambda_r$.

$$x_{mir}^* = \lambda_r x_m^* + (1 - \lambda_r)x_i, \text{ where } y_k^* = \widehat{y}_t \tag{2}$$

We feed the perturbed oracle sample to the classification model $f(\cdot)$ and get the model's prediction scores, $O_{mir}^* = f(x_{mir}^*) \in \mathbb{R}^K$. Then, we average the perturbed oracle samples' model outputs, to get $\bar{O}_{ir}^* = \frac{1}{M}\sum_{m=1}^M O_{mir}^*$. Finally, we compute the perturbed oracle samples' OOD score, $s_{ir}^* \in \mathbb{R}$, with an arbitrary output-based OOD scoring function $h(\cdot)$ such as MSP and MLS, i.e., $s_{ir}^* = h(\bar{O}_{ir}^*) \in \mathbb{R}$.

**Target-side perturbation** We perturb the target sample $x_t$ with the same auxiliary samples $\{x_i\}_{i=1}^N$, as $x_{ir} = \lambda_r x_t + (1 - \lambda_r)x_i$, and compute the OOD scores of the perturbed target sample as follows:

$$O_{ir} = f(x_{ir}) \in \mathbb{R}^K, \quad s_{ir} = h(O_{ir}) \in \mathbb{R} \tag{3}$$

**Comparison of perturbed samples' outputs** From perturbed target's and oracles' uncertainty scores, $(s_{ir}^*, s_{ir})$, we calculate the MixDiff score for the target sample, $x_t$, as shown in Equation 4. We average the differences between the two scores over the auxiliary samples and the Mixup ratios. This measures the model's uncertainty score of the target sample relative to similar ID samples when both undergo the same Mixup operation with an auxiliary sample $x_i$. We describe the overall procedure of MixDiff in Algorithm 1 and Figure 2.

$$\text{MixDiff} = \frac{1}{RN} \sum_{r=1}^{R} \sum_{i=1}^{N} (s_{ir} - s_{ir}^*) \tag{4}$$

As a final step, we multiply the MixDiff score by a scaling hyperparameter $\gamma$ and simply add this to the base OOD score of the target sample, $h(x_t)$, to inject additional information gained from the perturb-and-compare approach.

**Practical implementation** We note that the oracle-side procedure can be precomputed since it does not depend on the target sample. The target-side computations can be effectively parallelized since each perturbed target sample can be processed by the model, independent of the others. In our implementation, we organize the perturbed target samples in a single batch. Further speedup can be gained in remote API environments as API calls are often handled by multiple nodes. This scenario is similar to Ning et al. (2023), where concurrent API calls are employed to enhance latency in remote API environments.

---

**Algorithm 1** Computation of MixDiff Score

**Require**: target sample $x_t$, set of auxiliary samples $\{x_i\}_{i=1}^{N}$, set of Mixup rates $\{\lambda_r\}_{r=1}^{R}$, set of oracle samples for all $K$ classes $\{\Omega_k\}_{k=1}^{K}$ where $\Omega_k = \{(x_m^*, y_k^*)\}_{m=1}^{M}$, classifier model $f(\cdot)$, OOD scoring function $h(\cdot)$

1: $O_t = f(x_t)$
2: $\widehat{y}_t = \arg\max(O_t)$
3: $\{(x_m^*, y_k^*)\}_{m=1}^{M} \leftarrow \Omega_k$, where $y_k^* = \widehat{y}_t$
4: **for** $i \in \{1, \ldots, N\}$ **do**
5:     **for** $r \in \{1, \ldots, R\}$ **do**
6:        **for** $m \in \{1, \ldots, M\}$ **do**
7:           $O_{mir}^* \leftarrow f(\lambda_r x_m^* + (1 - \lambda_r)x_i)$
8:        **end for**
9:        $s_{ir}^* \leftarrow h\left(\frac{1}{M}\sum_{m=1}^{M} O_{mir}^*\right)$
10:        $O_{ir} \leftarrow f(\lambda_r x_t + (1 - \lambda_r)x_i)$
11:        $s_{ir} \leftarrow h(O_{ir})$
12:     **end for**
13: **end for**
14: $\text{MixDiff} \leftarrow \frac{1}{RN}\sum_{r=1}^{R}\sum_{i=1}^{N}(s_{ir} - s_{ir}^*)$

---

### 3.1 THEORETICAL ANALYSIS

To better understand how and when our method ensures performance improvements, we present a theoretical analysis of MixDiff. We use a similar theoretical approach to Zhang et al. (2021), but towards a distinct direction for analyzing a post hoc OOD scoring function. Proposition 1 reveals the decomposition of the OOD score function into two components: the OOD score of the unmixed clean target sample and the supplementary signals introduced by Mixup.

**Proposition 1** (OOD scores for mixed samples). *Let pre-trained model $f(\cdot)$ and base OOD score function $h(\cdot)$ be twice-differentiable functions. and $x_{i\lambda} = \lambda x_t + (1 - \lambda)x_i$ be a mixed sample with ratio $\lambda \in (0, 1)$. Then OOD score function of mixed sample, $h(f(x_{i\lambda}))$, is written as:*

$$h(f(x_{i\lambda})) = h(f(x_t)) + \sum_{l=1}^{3} \omega_l(x_t, x_i) + \varphi_t(\lambda)(\lambda - 1)^2, \tag{5}$$

*where $\lim_{\lambda \to 1} \varphi_t(\lambda) = 0$,*

$$\omega_1(x_t, x_i) = (\lambda - 1)(x_t - x_i)^T f'(x_t) h'(f(x_t))$$

$$\omega_2(x_t, x_i) = \frac{(\lambda - 1)^2}{2}(x_t - x_i)^T f''(x_t)(x_t - x_i) h'(f(x_t))$$

$$\omega_3(x_t, x_i) = \frac{(\lambda - 1)^2}{2}(x_t - x_i)^T f'(x_t)(x_t - x_i)^T f'(x_t) h''(f(x_t)).$$

We analyze MixDiff using the quadratic approximation of $h(f(x_{i\lambda}))$, omitting the higher order terms denoted as $\varphi_t(\lambda)$ in Equation 5. We verify how well the sum of the OOD score of the pure sample and omega approximates the OOD score of the mixed sample in Equation 5. As shown in Figure 3a, with a larger $\lambda$, approximation errors are smaller, as $\omega(x_t, x_i) = \sum_{l=1}^{3} \omega_l(x_t, x_i)$ decreases. The result of this observation is that $\omega(x_t, x_i)$ represents the impact caused by Mixup since it increases as $\lambda$ decreases. Then the additional signals caused by Mixup are from the first and second derivative of $f(\cdot)$ and $h(\cdot)$ and the difference between the input sample and the auxiliary sample.

We argue that perturbing both target and oracle samples and then comparing the results of them can be beneficial for OOD detection. Through Theorem 1, we demonstrate the effect of our perturb-and-compare strategy, MixDiff, on a simple linear model setup.

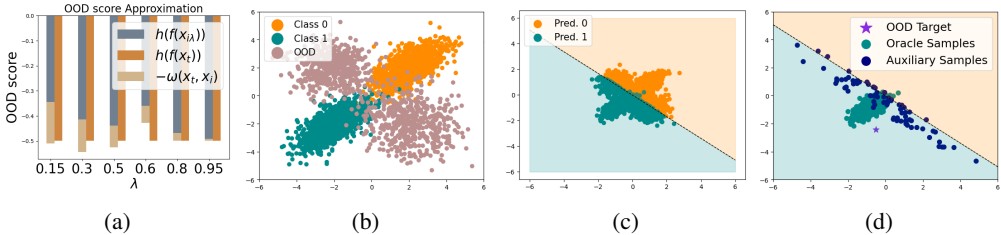

(a)  (b)  (c)  (d)

Figure 3: **(a)** Approximation error in Equation 5. Without higher-order terms, we can reasonably approximate the OOD score of mixed sample with decomposed terms. **(b)** Syntactic data distribution. Data is sampled from four independent Gaussian distributions, with two considered ID samples for each class and the other two as OOD samples. We train a logistic regression with this dataset. **(c)** The results of the prediction of the trained model. **(d)** Although the target sample is a hard OOD sample, there are auxiliary samples that guarantee MixDiff is positive under some reasonable conditions introduced in Theorem 1.

**Theorem 1.** *Let $h(x)$ and $f(x)$ be a MSP and linear model, $w^T x + b$, where $w$, $x \in \mathbb{R}^d$, and $b \in \mathbb{R}$, respectively. First we assume that the confidence of any arbitrary oracle sample $f(x_m)$ is sufficiently large, such that $0 < \frac{\sigma''(f_t)}{\sigma''(f_m)} < 1$. Additionally, we assume that $0 < f(x_m)f(x_t)$, $f(x_t) = f(x_m) + c$, where $c > 0$ (i.e., $x_t$ is one of the hard OOD samples to which the model have more confidence than corresponding oracle sample, $x_m$). For the binary classification task, there exists $x_i$ such that*

$$h(f(x_t)) - h(f(x_m)) + \sum_{l=1}^{3}(\omega_l(x_t, x_i) - \omega_l(x_m, x_i)) > 0. \tag{6}$$

In Theorem 1, we show that when the OOD sample is a high-confidence sample, i.e., a hard OOD sample, there exists an auxiliary sample $x_i$ that can be used to calibrate the overemphasis via MixDiff's perturb-and-compare mechanism. This provides a theoretical ground for our approach's effectiveness at discerning OOD samples that may not be detected by existing methods[1].

## 4 EXPERIMENTS

### 4.1 EXPERIMENTAL SETUP

We elaborate on the implementation details and present the descriptions on baselines. Other details on datasets and evaluation metrics are provided in Appendix E.

**Implementation details**  Following the zero-shot OOD detection approach (Esmaeilpour et al., 2022; Ming et al., 2022; Wang et al., 2023a) that utilizes pre-trained vision-language models' zero-shot classification capability, we employ CLIP ViT-B/32 model (Radford et al., 2021) as our classification model without any finetuning on ID samples. We construct the oracle set by randomly sampling $M$ samples per class from the train split of each dataset. For a given target sample, we simply use the other samples in the same batch as the auxiliary set. We note that in this simple setup, the auxiliary samples do not require labeling and may even contain OOD samples. Instead of searching hyperparameters for each dataset, we perform one hyperparameter search on Caltech101 (Fei-Fei et al., 2004) and use the same hyperparameters across all the other datasets, which is in line with a more realistic OOD detection setting (Liang et al., 2018) [2].

**Baselines**  We take MSP (Hendrycks & Gimpel, 2017), MLS (Hendrycks et al., 2022), energy score (Liu et al., 2020), Shannon entropy (Lakshminarayanan et al., 2017) and MCM (Ming et al., 2022) as the output-based training-free baselines. We also include methods that require extra training for comparison. ZOC (Esmaeilpour et al., 2022) is a zero-shot OOD detection method based on CLIP

---

[1]Proof and details of Proposition 1 and Theorem 1 are in Appendix B and C, respectively. We also show that Theorem 1 holds for MLS and Entropy in Appendix C.

[2]We provide more details of implementation in Appendix E.

| Method | Training-free | CIFAR10 | CIFAR100 | CIFAR+10 | CIFAR+50 | TinyImageNet | Avg. |
|---|---|---|---|---|---|---|---|
| CSI (Tack et al., 2020) | ✗ | 87.0±4.0 | 80.4±1.0 | 94.0±1.5 | 97.0 | 76.9±1.2 | 87.0 |
| CAC (Miller et al., 2021) | ✗ | 80.1±3.0 | 76.1±0.7 | 87.7±1.2 | 87.0 | 76.0±1.5 | 84.9 |
| CLIP+CAC (Miller et al., 2021) | ✗ | 89.3±2.0 | **83.5**±1.2 | 96.5±0.5 | 95.8 | **84.6**±1.7 | 89.9 |
| ZOC [†] (Esmaeilpour et al., 2022) | ✗ | 91.5±2.5 | 82.7±2.8 | 97.6±1.1 | 97.1 | 82.6±3.1 | 90.3 |
| MixDiff+ZOC [†] | ✗ | **92.2**±2.5 | 82.8±2.4 | **98.2**±1.2 | **98.5** | 82.9±3.3 | **90.9** |
| MSP (Hendrycks & Gimpel, 2017) | ✓ | 88.7±2.0 | 78.2±3.1 | 95.0±0.8 | 95.1 | 80.4±2.5 | 87.5 |
| MLS (Hendrycks et al., 2022) | ✓ | 87.8±3.0 | 80.0±3.1 | 96.1±0.8 | 96.0 | 84.0±1.2 | 88.8 |
| Energy (Liu et al., 2020) | ✓ | 85.4±3.0 | 77.6±3.7 | 94.9±0.9 | 94.8 | 83.2±1.2 | 87.2 |
| Entropy (Lakshminarayanan et al., 2017) | ✓ | 89.9±2.6 | 79.9±2.5 | 96.8±0.8 | 96.8 | 82.2±2.3 | 89.1 |
| MCM (Ming et al., 2022) | ✓ | 90.6±2.9 | 80.3±2.1 | 96.9±0.8 | 97.0 | 83.1±2.2 | 89.6 |
| MixDiff+MSP | ✓ | 89.2±1.6 | 80.1±2.8 | 96.7±0.8 | 96.9 | 81.6±2.6 | 88.9 |
| MixDiff+MLS | ✓ | 87.9±2.1 | 80.5±2.2 | 96.5±0.7 | 96.9 | **84.5**±0.9 | 89.3 |
| MixDiff+Energy | ✓ | 85.6±2.2 | 78.3±2.7 | 95.4±0.8 | 95.9 | 83.6±1.1 | 87.8 |
| MixDiff+Entropy | ✓ | 90.7±1.8 | 81.0±2.6 | 97.6±0.8 | 97.6 | 82.9±2.4 | 90.0 |
| MixDiff+MCM | ✓ | **91.4**±1.8 | **81.4**±2.6 | **97.5**±0.9 | **97.7** | 83.9±2.2 | **90.4** |

Table 1: Average AUROC scores for five datasets. The highest and second highest AUROC scores from each block are highlighted with **bold** and underline. The value on the right side of ± denotes the standard deviation induced from 5 different OOD, ID class splits. † indicates the evaluation setting described in Appendix E.4.

(Radford et al., 2021) that requires training a separate candidate OOD class name generator. CAC (Miller et al., 2021) relies on train-time loss function modification. CAC shows the best performance among the train-time modification methods that are compatible with CLIP. We take CAC trained with the same CLIP ViT-B/32 backbone as a baseline (CLIP+CAC) (Esmaeilpour et al., 2022).

## 4.2 LOGITS AS MODEL OUTPUTS

First, we assume a more lenient access constraint whereby logits are provided as the model $f(\cdot)$'s outputs. This setup facilitates validation of MixDiff's OOD score enhancement ability on both the logit-based and probability-based scores. Note that, in this setup, the perturbed oracle samples' probability-based OOD scores are computed after averaging out $M$ perturbed oracle samples in the logit-space, i.e., $\bar{O}_{ir}^* = \frac{1}{M} \sum_{m=1}^{M} O_{mir}^*$.

The results in Table 1 show that the perturb-and-compare approach is effective at enhancing output-based OOD scores, to a degree where one of the training-free methods, MixDiff+MCM, outperforming a training-based method CLIP+CAC (Miller et al., 2021). Equipping MixDiff with the best performing non-training-free method, ZOC (Esmaeilpour et al., 2022), also yields performance improvements[3]. Table 2 presents a comprehensive performance analysis of MixDiff in relation to other baselines, utilizing commonly employed metrics for OOD detection studies. Our findings show MixDiff can boost OOD detection performance, particularly in FPR95 and AUCPR as well as AUROC.

| Method | Training-free | AUROC (↑) | FPR95 (↓) | AUCPR (↑) |
|---|---|---|---|---|
| ZOC [†] | ✗ | 82.7±2.8 | **64.0**±6.9 | 94.1±1.0 |
| MixDiff+ZOC [†] | ✗ | 82.8±2.4 | 65.2±12.0 | **95.0**±0.7 |
| MSP | ✓ | 78.2±3.1 | 60.4±5.3 | 91.4±1.9 |
| MLS | ✓ | 80.0±3.1 | 62.3±5.2 | 92.9±1.6 |
| Energy | ✓ | 77.6±3.7 | 65.4±4.2 | 91.9±1.9 |
| Entropy | ✓ | 79.9±2.5 | 58.8±5.2 | 92.0±1.7 |
| MixDiff+MSP | ✓ | 80.1±2.8 | 60.1±4.8 | 92.3±1.5 |
| MixDiff+MLS | ✓ | 80.5±2.2 | 62.5±4.1 | **92.9**±1.2 |
| MixDiff+Energy | ✓ | 78.3±2.7 | 65.9±3.4 | 92.1±1.4 |
| MixDiff+Entropy | ✓ | **81.0**±2.6 | **58.4**±4.8 | 92.6±1.5 |

Table 2: Performance comparison with various metrics.

## 4.3 PREDICTION PROBABILITIES AS MODEL OUTPUTS

We now take a more restricted environment where the only accessible part of the model is its output prediction probabilities. To the best of our knowledge, none of the existing OOD score enhancement methods are applicable in this environment. Logits are required in the case of Softmax temperature scaling (Liang et al., 2018). ODIN's gradient-based input preprocessing (Liang et al., 2018) or weight pruning methods (Sun & Li, 2022) assume an access to the model's parameters. The model's internal activations are required in the case of activation clipping (Sun et al., 2021) and activation pruning (Djurisic et al., 2023). We take a linear combination of entropy and MSP scores with a scaling hyperparameter tuned on the Caltech101 dataset as a baseline (MSP+Entropy). The re-

---

[3]We provide details on adaptation of MixDiff with ZOC in Appendix E.

| Method | Access | CIFAR10 | CIFAR100 | CIFAR+10 | CIFAR+50 | TinyImageNet | Avg. |
|---|---|---|---|---|---|---|---|
| MSP (Hendrycks & Gimpel, 2017) | Prediction prob. | $88.7_{\pm2.0}$ | $78.2_{\pm3.1}$ | $95.0_{\pm0.8}$ | 95.1 | $80.4_{\pm2.5}$ | 87.5 |
| Entropy (Lakshminarayanan et al., 2017) | Prediction prob. | $89.9_{\pm2.6}$ | $79.9_{\pm2.5}$ | $96.8_{\pm0.8}$ | 96.8 | $82.2_{\pm2.3}$ | 89.1 |
| MSP+Entropy | Prediction prob. | $89.9_{\pm2.6}$ | $79.9_{\pm2.5}$ | $96.8_{\pm0.8}$ | 96.8 | $82.2_{\pm2.3}$ | 89.1 |
| MixDiff+MSP (Prediction probabilities) | Prediction prob. | $89.4_{\pm1.3}$ | $80.0_{\pm2.8}$ | $96.5_{\pm0.8}$ | 96.8 | $81.8_{\pm2.4}$ | 88.9 |
| MixDiff+Entropy (Prediction probabilities) | Prediction prob. | $\mathbf{91.1}_{\pm1.6}$ | $\underline{80.9}_{\pm2.6}$ | $\underline{97.1}_{\pm0.8}$ | $\underline{97.3}$ | $\underline{82.9}_{\pm2.3}$ | $\mathbf{89.9}$ |
| with unlabeled oracle | Prediction prob. | $\underline{91.0}_{\pm1.6}$ | $80.5_{\pm2.9}$ | $\underline{97.1}_{\pm0.8}$ | $\underline{97.3}$ | $82.7_{\pm2.1}$ | $\underline{89.7}$ |
| with oracle as auxiliary | Prediction prob. | $90.6_{\pm1.7}$ | $\mathbf{81.1}_{\pm2.0}$ | $\mathbf{97.3}_{\pm0.7}$ | $\mathbf{97.4}$ | $82.9_{\pm2.2}$ | $\mathbf{89.9}$ |
| with random ID as auxiliary | Prediction prob. | $90.8_{\pm1.5}$ | $\mathbf{81.1}_{\pm2.1}$ | $96.8_{\pm1.0}$ | 96.8 | $\underline{82.9}_{\pm2.3}$ | $\underline{89.7}$ |
| with perturb only | Prediction prob. | $89.4_{\pm2.9}$ | $79.5_{\pm2.7}$ | $\underline{97.1}_{\pm0.9}$ | 97.2 | $81.6_{\pm2.5}$ | 89.0 |
| with random ID as oracle | Prediction prob. | $89.5_{\pm2.8}$ | $79.6_{\pm2.7}$ | $\underline{97.1}_{\pm0.9}$ | $\underline{97.3}$ | $81.7_{\pm2.5}$ | 89.0 |
| Random score from uniform distribution | Prediction label | $49.6_{\pm0.5}$ | $49.8_{\pm1.1}$ | $49.8_{\pm0.7}$ | 50.1 | $49.8_{\pm0.4}$ | 49.8 |
| MixDiff with random ID as auxiliary | Prediction label | $62.4_{\pm4.1}$ | $59.4_{\pm6.2}$ | $65.6_{\pm1.5}$ | 65.4 | $63.3_{\pm2.8}$ | 63.2 |
| MixDiff with oracle as auxiliary | Prediction label | $61.9_{\pm3.7}$ | $55.1_{\pm7.1}$ | $59.9_{\pm1.1}$ | 59.8 | $55.6_{\pm2.7}$ | 58.4 |
| DML (Zhang & Xiang, 2023) | Activation | $87.8_{\pm3.0}$ | $80.0_{\pm3.1}$ | $96.1_{\pm0.8}$ | 96.0 | $\mathbf{84.0}_{\pm1.2}$ | 88.8 |
| ASH (Djurisic et al., 2023) | Activation | $85.2_{\pm3.8}$ | $75.4_{\pm4.4}$ | $92.5_{\pm0.9}$ | 92.4 | $77.2_{\pm3.1}$ | 84.5 |

Table 3: AUROC scores on more restricted access scenarios. The upper, middle blocks contain results on the environment where only the prediction probabilities, prediction labels are available, respectively. The lower block's methods require models' inner activations

sults are presented in Table 3. Even in this constrained situation, MixDiff can effectively enhance output-based OOD scores by utilizing additional information gained from the perturb-and-compare approach, while MSP score fails to provide entropy score any meaningful performance gain.

**Ablations** We present the ablation results of MixDiff framework in Table 3 to illuminate each component's effect on performance. We take MixDiff+Entropy for these experiments. First, we progressively increase the homogeneity of auxiliary samples by changing the in-batch auxiliary samples, which may contain OOD samples, to random ID samples (random ID as auxiliary), and to the other oracle samples with the same predicted label as the target (oracle as auxiliary). Removal of the comparison part of the perturb-and-compare approach by adding only the perturbed target's scores without comparing with the perturbed oracles' scores results in performance degradation (perturb only). On the oracle side, randomly choosing oracle samples instead of finding similar oracle samples using the predicted class label leads to decreased performance (random ID as oracle). We show that utilization of pseudo-oracle is possible by selecting top-$M$ most similar samples from $M \times K$ unlabeled ID samples with similarity calculated from the dot product of the prediction probabilities of the target and the unlabeled oracle samples (unlabeled oracle).

## 4.4 PREDICTION LABELS AS MODEL OUTPUTS

We push the limits of the model access by assuming that only the predicted class labels are available without any scores attached to them. We apply MixDiff by representing the model's predictions as one-hot vectors and taking the difference between the perturbed target's predicted label and the corresponding perturbed oracles' average score for that label in Equation 4. As there is no base OOD score applicable in the environment, we use the MixDiff score alone. The results in Table 3 show that MixDiff is applicable even in this extremely constrained access environment.

## 4.5 ROBUSTNESS TO ADVERSARIAL ATTACKS

In adversarial attack of an OOD detector, the attacker creates a small, indistinguishable modification to a target sample with the purpose of increasing (decreasing) the model's confidence for a given OOD (ID) sample (Chen et al., 2022a; 2021; Aziz-malayeri et al., 2022). These modifications can be viewed as injection of certain artificial features, specifically designed to induce more confident or uncertain outputs from the

| Method | CIFAR10 | | | | CIFAR100 | | | |
|---|---|---|---|---|---|---|---|---|
| | Clean | In | Out | Both | Clean | In | Out | Both |
| Entropy | $\underline{89.88}$ | 47.42 | 13.77 | 2.678 | $\underline{79.87}$ | 36.86 | 14.38 | 2.21 |
| MixDiff+Entropy | $\mathbf{90.64}$ | $\underline{54.71}$ | $\underline{31.77}$ | 9.084 | $\mathbf{81.11}$ | $\underline{47.42}$ | $\underline{31.40}$ | $\underline{9.08}$ |
| MixDiff Only | 88.16 | $\mathbf{61.00}$ | $\mathbf{40.28}$ | $\mathbf{20.45}$ | 78.05 | $\mathbf{58.84}$ | $\mathbf{44.19}$ | $\mathbf{27.48}$ |

Table 4: AUROC scores on various attack scenarios. "In"/"Out" indicates all of the ID/OOD samples are adversarially modified. "Both" indicates all of the ID, OOD samples are adversarially modified. MixDiff Only refers to the score in Equation 4.

model. Our motivation in Section 1 suggests that these artificial features may also be less robust to perturbations. We test this by evaluating MixDiff under adversarial attack. The results in Table

4 indicate that the contributing features that induce ID/OOD misclassification are less robust to perturbations and that MixDiff can effectively exploit such brittleness. Detailed description of the experimental setup is in Appendix E.6.

### 4.6 EXPERIMENTS ON OUT-OF-SCOPE DETECTION TASK

**Out-of-scope detection**  We take the MixDiff framework to out-of-scope (OOS) detection task to check its versatility in regard to the modality of the input. To reliably fulfill users' queries or instructions, understanding the intent behind a user's utterance forms a crucial aspect of dialogue systems. In the intent classification task, models are tasked to extract the intent behind a user utterance. While there has been an inflow of development in the area for the improvement of classification performance (Kumar et al., 2022; Zhang et al., 2022), there is no guarantee that a given query's intent is in the set of intents that the model is able to classify, especially in a real-world setting. Out-of-scope detection task (Chen & Yu, 2021; Zhan et al., 2021) concerns with detection of such user utterances, so that it would refrain from performing an unintended action when the intent belongs to none of the intents listed in an intent classifier.

**MixDiff with textual input**  Unlike images whose continuousness lends itself to a simple Mixup operation, the discreteness of texts renders Mixup of texts not as straightforward. While there are several works that explore Mixup of text, most of these require access to the model parameters (Yoon et al., 2021; Kong et al., 2022; Guo et al., 2019). This limits the MixDiff framework's applicability in an environment where the model is served as an API (Sun et al., 2022b;a), which is becoming more and more prevalent with the rapid development of large language models (Brown et al., 2020; OpenAI, 2023; Touvron et al., 2023). Following this trend, we assume a more challenging environment with the requirement that Mixup be performed on the input level. To this end, we simply concatenate the text pair and let Mixup happen while the pair is inside the model (Hao et al., 2023).

**Experimental setup**  We run OOS detection experiments using 4 intent classification datasets: CLINC150 (Larson et al., 2019), Banking77 (Casanueva et al., 2020), ACID (Acharya & Fung, 2020), TOP (Gupta et al., 2018). Following Zhan et al. (2021); Lin & Xu (2019), we randomly split the provided classes into in-scope and OOS intents, with in-scope intent class ratios of 25%, 50%, 75%. For the intent classification model, we finetune the BERT-base model (Devlin et al.,

| Method | CLINC150 | Banking77 | ACID | TOP | Average |
|---|---|---|---|---|---|
| MSP | 93.02 | 85.43 | 88.98 | 90.01 | 89.36 |
| MLS | 93.56 | 85.02 | 88.91 | 90.06 | 89.39 |
| Energy | 93.61 | 84.99 | 88.83 | 90.06 | 89.37 |
| Entropy | 93.29 | 85.59 | 88.87 | 90.02 | 89.44 |
| MixDiff+MSP | 93.42 | 85.75 | 89.18 | **90.68** | 89.76 |
| MixDiff+MLS | 93.88 | 85.46 | **89.24** | 90.35 | 89.73 |
| MixDiff+Energy | **93.89** | 85.51 | 89.18 | 90.35 | 89.73 |
| MixDiff+Entropy | 93.67 | **85.98** | 89.13 | **90.68** | **89.87** |

Table 5: Average AUROC scores for out-of-scope detection task.

2019) on the in-scope split of each dataset's train set. For each in-scope ratio, we construct 10 in-scope, OOS splits with different random seeds and evaluate the test set containing the entirety of the intent classes.

**Results**  We report the average AUROC scores in Table 5, each of which is averaged over the in-scope class ratios as well as the class splits. Even with a simple Mixup method that simply concatenates the text pair, MixDiff consistently improves the performance across diverse datasets. The results suggest that the MixDiff framework's applicability is not limited to images and that the framework can be applied to other modalities with an appropriate perturbation method.

## 5 CONCLUSION

In this paper, we present a new OOD detection framework, MixDiff, that boosts OOD detection performance in constrained access scenarios. MixDiff is based on the perturb-and-compare approach that measures how the model's confidence in the target sample behaves compared to a similar ID sample when both undergo an identical perturbation. This provides an additional signal that cannot be gained from the limited information of the target sample's model output alone. We provide theoretical grounds for the framework's effectiveness and empirically validate our approach on multiple degrees of restricted access scenarios, the most extreme of which is where only the model's prediction labels are available. Our experimental results show that MixDiff is an effective OOD detection method for constrained access scenarios where the applicability of existing methods is limited.

# 6 REPRODUCIBILITY

We make an anonymized version of our code that we have used to run the experiments in the paper available. The implementation for the experiments on the more restricted access scenarios where only the models' prediction probabilities or class labels were accessible, is made available at: https://anonymous.4open.science/r/mixdiff_hard. The implementation for the experiments where logits were available as models' prediction scores is available at: https://anonymous.4open.science/r/mixdiff_easy. The latter includes the out-of-scope detection experiments as well as output-based OOD score baselines.

# 7 ETHICS

We note that the pre-trained models, namely, CLIP (Devlin et al., 2019) and BERT (Radford et al., 2021), that we have used in our experiments may contain social biases as these models are trained on large amounts of unfiltered data on the Web. How these biases manifest when deciding whether a sample is OOD or ID is crucial for fairness, since it is possible for these biases to negatively affect the model's decision process. Future work may explore how the models' biases may affect MixDiff's OOD detection process and selection of oracle or auxiliary samples that induce unbiased out-of-distribution detection.

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

## A  NOTATION

| Notation | Definition |
|---|---|
| $f(\cdot)$ | Classifier model. |
| $h(\cdot)$ | Arbitrary output-based OOD score function. |
| $M$ | The number of oracle samples of each class. |
| $R$ | The number of Mixup ratios. |
| $N$ | The number of auxiliary samples that will be mixed with the oracle or target samples. |
| $\Omega_k$ | Set of oracle sample and label pairs for the $k$-th class. |
| $\Omega$ | Set of oracle sample and label pairs of all classes, $\{\Omega_k\}_{k=1}^{K}$. |
| $\lambda_r$ | $r$-th Mixup ratio. |
| $x_t$ | The target sample. |
| $x_{ir}$ | Mixed sample from the target $x_t$ and $i$-th auxiliary sample with Mixup ratio of $\lambda_r$. |
| $x_{mir}^{*}$ | Mixed sample from the $m$-th oracle sample $i$-th auxiliary sample with Mixup ratio of $\lambda_r$. |
| $O_{ir}$ | The prediction scores from the mixture of the target and $i$-th auxiliary sample with the Mixup ratio $\lambda_r$. |
| $O_{mir}^{*}$ | The prediction scores from the mixture of the $m$-th oracle and $i$-th auxiliary sample with the Mixup ratio $\lambda_r$. |
| $\bar{O}_{ir}^{*}$ | The mean of $\{O_{mir}^{*}\}_{m=1}^{M}$ along the subscript $m$. |
| $s_{ir}$ | OOD score induced by $O_{ir}$. |
| $s_{ir}^{*}$ | OOD score induced by $\bar{O}_{ir}^{*}$. |
| $\gamma$ | The scaling hyperparameter to which the MixDiff score will be multiplied. |

## B  PROOF OF PROPOSITION 1

**Proposition 1** (OOD score function for mixed samples). *Let pre-trained model $f(\cdot)$ and base OOD score function $h(\cdot)$ be a twice-differentiable function. and $x_{i\lambda} = \lambda x_t + (1-\lambda)x_i$ be a mixed sample with ratio $\lambda \in (0,1)$. Then base OOD score function of mixed sample, $h(f(x_{i\lambda}))$, is written as:*

$$h(f(x_{i\lambda})) = h(f(x_t)) + \sum_{l=1}^{3} \omega_l(x_t, x_i) + \varphi_t(\lambda)(\lambda - 1)^2 \tag{B.1}$$

*where* $\lim_{\lambda \to 1} \varphi_t(\lambda) = 0$,

$$\omega_1(x_t, x_i) = (\lambda - 1)(x_t - x_i)^T f'(x_t) h'(f(x_t))$$

$$\omega_2(x_t, x_i) = \frac{(\lambda - 1)^2}{2}(x_t - x_i)^T f''(x_t)(x_t - x_i) h'(f(x_t))$$

$$\omega_3(x_t, x_i) = \frac{(\lambda - 1)^2}{2}(x_t - x_i)^T f'(x_t)(x_t - x_i)^T f'(x_t) h''(f(x_t)).$$

*Proof of Proposition 1.* Let $\psi_t(\lambda) = h(f(x_{i\lambda}))$ which is modified function of $h(f(x_{i\lambda}))$ having $\lambda$ as an input. If $h(\cdot)$ and $f(\cdot)$ are twice differentiable w.r.t each input. By the second-order Taylor approximation,

$$\psi_t(\lambda) = \psi_t(1) + \psi_t'(1)(\lambda - 1) + \frac{1}{2}\psi_t''(1)(\lambda - 1)^2 + \varphi_t(\lambda)(\lambda - 1)^2, \tag{B.2}$$

where $\lim_{\lambda \to 1} \varphi_t(\lambda) = 0$.

$$\psi_t'(\lambda) = \frac{\partial x_{i\lambda}}{\partial \lambda} \frac{\partial f(x_{i\lambda})}{\partial x_{i\lambda}} \frac{\partial h(f(x_{i\lambda}))}{\partial f(x_{i\lambda})} = (x_t - x_i)^T f'(x_{i\lambda}) h'(f(x_{i\lambda}))$$

Since $\frac{\partial}{\partial \lambda}(x_t - x_i)^T f'(x_{i\lambda}) h'(f(x_{i\lambda})) = \frac{\partial}{\partial \lambda}[(x_t - x_i)^T f'(x_{i\lambda})] h'(f(x_{i\lambda})) + (x_t - x_i)^T f'(x_{i\lambda}) \frac{\partial}{\partial \lambda}[h'(f(x_{i\lambda}))]$ and $\frac{\partial}{\partial \lambda}(x_t - x_i)^T f'(x_{i\lambda}) = (x_t - x_i)^T f''(x_{i\lambda})(x_t - x_i)$,

$$\psi_t''(\lambda) = (x_t - x_i)^T f''(x_{i\lambda})(x_t - x_i) h'(f(x_{i\lambda})) + (x_t - x_i)^T f'(x_{i\lambda})(x_t - x_i)^T f'(x_{i\lambda}) h''(f(x_{i\lambda}))$$

When $\lambda = 1$

$$\psi_t'(1) = (x_t - x_i)^T f'(x_t) h'(f(x_t))$$

$$\psi_t''(1) = (x_t - x_i)^T f''(x_t)(x_t - x_i) h'(f(x_t)) + (x_t - x_i)^T f'(x_t)(x_t - x_i)^T f'(x_t) h''(f(x_t))$$

Fianlly, we derive Equation B.1 in proposition 1 as

$$h(f(x_{i\lambda})) = h(f(x_t)) + (\lambda - 1)(x_t - x_i)^T f'(x_t) h'(f(x_t)) \tag{B.3}$$

$$+ \frac{(\lambda - 1)^2}{2}(x_t - x_i)^T f''(x_t)(x_t - x_i) h'(f(x_t)) \tag{B.4}$$

$$+ \frac{(\lambda - 1)^2}{2}(x_t - x_i)^T f'(x_t)(x_t - x_i)^T f'(x_t) h''(f(x_t)) \tag{B.5}$$

$$+ \varphi_t(\lambda)(\lambda - 1)^2.$$

$$\square$$

## C  PROOF OF THEOREM 1

**Theorem 1.** *Let h(x) and f(x) be a MSP and linear model, $w^T x + b$, where $w$, $x \in \mathbb{R}^d$, and $b \in \mathbb{R}$, respectively. First we assume that the confidence of any arbitrary oracle sample $f(x_m)$ is sufficiently large, such that $0 < \frac{\sigma''(f_t)}{\sigma''(f_m)} < 1$. Additionally, we assume that $0 < f(x_m)f(x_t)$, $f(x_t) = f(x_m) + c$, where $c > 0$ (i.e., $x_t$ is one of the hard OOD samples to which the model have more confidence than corresponding oracle sample, $x_m$). For the binary classification task, there exists $x_i$ such that*

$$h(f(x_t)) - h(f(x_m)) + \sum_{l=1}^{3}(\omega_l(x_t, x_i) - \omega_l(x_m, x_i)) > 0. \tag{C.6}$$

*Proof of Theorem 1.* Considering MSP in binary classification task, MSP $= -\max(\sigma(f(x)), 1 - \sigma(f(x)))$. $f'(x) = w$, $f''(x) = \mathbf{0}$,

$$h'(f(x)) = \begin{cases} -\sigma'(f(x)) & \text{if } f(x) > 0 \\ \sigma'(f(x)) & \text{otherwise} \end{cases}$$

$$h''(f(x)) = \begin{cases} -\sigma''(f(x)) & \text{if } f(x) > 0 \\ \sigma''(f(x)) & \text{otherwise} \end{cases}$$

$\sigma(\cdot)$ denotes the sigmoid function. Assuming the target input is one of hard OOD samples that can be interpreted as $f(x_t) = f(x_m) + c$ and $h(f(x_t)) < h(f(x_m))$ where $0 < f(x_m) < f(x_t), 0 < c$ and $0.5 < \sigma(f(x_m)) < \sigma(f(x_t))$. Then $h(f(x_t)) - h(f(x_m)) = -\sigma(f(x_t)) + \sigma(f(x_m))$. $-0.5 < -\sigma(f(x_t)) + \sigma(f(x_m)) < 0$.

Equation C.6 is equivalent to Equation C.7 as $\omega_2 = 0$ under the assumption that $f(x)$ is a linear model.

$$h(f(x_t)) - h(f(x_m)) + (\omega_1(x_t, x_i) - \omega_1(x_m, x_i)) + (\omega_3(x_t, x_i) - \omega_3(x_m, x_i)) > 0 \tag{C.7}$$

$$\omega_1(x_t, x_i) - \omega_1(x_m, x_i) = (\lambda - 1)[(x_t - x_i)^T w(-\sigma'(f(x_t))) - (x_m - x_i)^T w(-\sigma'(f(x_m)))] \tag{C.8}$$

$$= (\lambda - 1)[(f(x_i) - f(x_t))\sigma'(f(x_t)) - (f(x_i) - f(x_m))\sigma'(f(x_m))] \tag{C.9}$$

$$= (\lambda - 1)[(f(x_i) - f(x_t))\sigma'(f(x_t)) - (f(x_i) - f(x_t) + c)\sigma'(f(x_m))] \tag{C.10}$$

$$= (\lambda - 1)[(f(x_i) - f(x_t))(\sigma'(f(x_t)) - \sigma'(f(x_m))) - c\sigma'(f(x_m))]. \tag{C.11}$$

Because we assume $0 < f(x_m) < f(x_t)$, $\sigma'(f(x_t)) - \sigma'(f(x_m)) < 0$, and $0 < \lambda < 1$, $(\lambda - 1)(f(x_i) - f(x_t))(\sigma'(f(x_t)) - \sigma'(f(x_m))) \geq 0$ when $f(x_i) - f(x_t) \geq 0$. When $(\omega_1(x_i, x_i) - \omega_1(x_m, x_i)) \geq 0$,

$$f(x_i) \geq f(x_t) + \frac{c\sigma'(f(x_m))}{\sigma'(f(x_t)) - \sigma'(f(x_m))}. \tag{C.12}$$

$f(x_i)$ denotes the confidence of the model with respect to auxiliary sample $x_i$. When $f(x_i)$ satisfies the above condition, Equation C.7 holds when

$$\omega_3(x_t, x_i) - \omega_3(x_m, x_i) \geq 0.5. \tag{C.13}$$

Let $0 < \tau = \frac{h''(f(x_t))}{h''(f(x_m))} < 1$, then

$$[(x_t - x_i)^T w]^2 h''(f(x_t)) - [(x_m - x_i)^T w]^2 h''(f(x_m)) \leq 0.5. \tag{C.14}$$

$$[(f(x_t) - f(x_i))^2 \tau - (f(x_m) - f(x_i))^2] h''(f(x_m)) \leq 0.5. \tag{C.15}$$

Because of $h''(f(x_t)) < 0$,

$$(f(x_t) - f(x_i))^2 \tau - (f(x_m) - f(x_i))^2 \leq \frac{0.5}{h''(f(x_t))} \tag{C.16}$$

$$(f(x_t) - f(x_i))^2 \tau - (f(x_t) - c - f(x_i))^2 \leq \frac{0.5}{h''(f(x_t))}. \tag{C.17}$$

Let $t = f(x_t) - f(x_i)$, then

$$t^2 \tau - (t - c)^2 = (\tau - 1)t^2 + 2ct - c^2. \tag{C.18}$$

By reformulating the Equation C.18 with respect to $f(x_i)$, we obtain the following expression.

$$(\tau - 1)f(x_i)^2 - 2((\tau - 1)f(x_t) + c)f(x_i) + (\tau - 1)f(x_t)^2 + 2cf(x_t) - c^2 - 2\sigma''(f(x_m)) \leq 0. \tag{C.19}$$

When $0 < \tau < 1$, the Equation C.19 is a concave quadratic function with respect to $f(x_i)$ and the discriminant of the Equation C.19 with respect to $f(x_i)$ is positive and The value of the right side of C.12 exists between the two solution values for which C.19 equals zero with respect to $f(x_i)$. $\square$

**Lemma 1.** *Considering* Entropy *OOD score function in binary classification task,* Entropy $= -(\sigma(f(x))\log(\sigma(f(x))) + (1 - \sigma(f(x)))\log(1 - \sigma(f(x))))$. $f'(x) = w$ and $f''(x) = \mathbf{0}$. *Let us assume prediction scores of a hard OOD sample and an oracle sample are* $f(x_t), f(x_m) > 0, f(x_t) = f(x_m) + c, c > 0$, *then* $-\epsilon < h(f(x_t)) - h(f(x_m)) < 0$, *where* $-\epsilon < 0$ *denotes the lower bound of the difference between the OOD scores of the target and oracle samples. Followed by Equation C.11,*

$$(\lambda - 1)(f(x_t) - f(x_m))(h'(f(x_t)) - h'(f(x_m))) + (\lambda - 1)ch'(f(x_m)).$$

*Because the sign of* $h'(f(x_m))$ *is a negative when* $f(x_m) > 0$, $(\lambda - 1)ch'(f(x_m)) \geq 0$. $h(f(x_t)) - h(f(x_m)) + (\omega_1(x_t, x_i) - \omega_1(x_m, x_i)) \geq 0$, *where* $(\lambda - 1)(f(x_t) - f(x_m))(h'(f(x_t)) - h'(f(x_m))) \geq \epsilon$.

$$f(x_i) \geq f(x_t) - \frac{\epsilon}{(\lambda - 1)(h'(f(x_t)) - h'(f(x_m)))}, \text{ if } h'(f(x_t)) - h'(f(x_m)) > 0 \tag{C.20}$$

$$f(x_i) \leq f(x_t) - \frac{\epsilon}{(\lambda - 1)(h'(f(x_t)) - h'(f(x_m)))}, \text{ if } h'(f(x_t)) - h'(f(x_m)) < 0 \tag{C.21}$$

*Under the assumption that* $f(x_i)$ *satisfies the above condition, Equation C.6 holds when*

$$\omega_3(x_t, x_i) - \omega_3(x_m, x_i) \geq 0. \tag{C.22}$$

*Let* $\tau = \frac{h''(f(x_m))}{h''(f(x_t))} > 0$, *then we follow the same step in Equation C.15 - Equation C.18. There exists* $x_i$ *such that it satisfies Equation C.22 and Equation C.21.*

**Lemma 2.** *Considering* MLS *OOD score function in binary classification task,* MLS *= -f(x). Equation C.6 is equivalent to Equation C.23 as* $\omega_2 = \omega_3 = 0$ *because* $f''(x) = h''(f(x)) = 0$.

$$h(f(x_t)) - h(f(x_m)) + \omega_1(x_t, x_i) - \omega_1(x_m, x_i) > 0 \tag{C.23}$$

*The right hand-side of Equation C.23 is written as*

$$-f(x_t) + f(x_m) + (1 - \lambda)[(f(x_t) - f(x_i)) - (f(x_m) - f(x_i))] \tag{C.24}$$

$$= -f(x_t) + f(x_m) + (1 - \lambda)(f(x_t) - f(x_m)) \tag{C.25}$$

$$= -\lambda f(x_t) + \lambda f(x_m). \tag{C.26}$$

*If* $x_t$ *is an OOD sample and* $f(x_m) > f(x_t)$ *where* $f(x_t), f(x_m) > 0$, *Equation C.23 holds.*

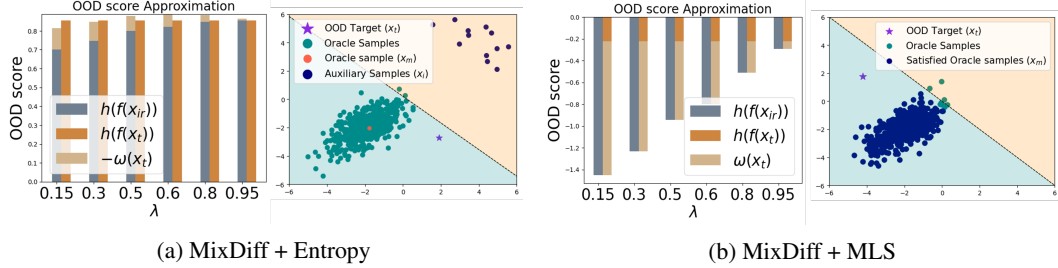

(a) MixDiff + Entropy         (b) MixDiff + MLS

Figure 4: The first figure in (a) and (b) compares the OOD score between the mixed target sample's OOD score and the approximated OOD score. The second figure in (a) and (b) shows the example of auxiliary samples and oracle samples that satisfy the condition to guarantee that MixDiff is positive.

## D    EXPERIMENTAL VALIDATION OF PROPOSITION 1 AND THEOREM 1

We experimentally ensure the proposition 1 and theorem 1 when the base OOD score function is Entropy and MLS, respectively. We verify the theorem 1 on the synthetic dataset consisting of 2-dimensional features. In the second figure in Figure 4a, we plot the OOD target sample, an oracle sample that has the same class as the prediction of the target sample, and auxiliary samples that satisfy the condition that makes MixDiff positive. In the second figure in Figure 4b, We show only the target sample, the oracle samples, and the oracle samples that meet the condition for having a positive MixDiff. Due to the assumption of binary classification with a linear model, it eliminates the effect of auxiliary samples.

## E    EXPERIMENTAL DETAILS

### E.1    EXPERIMENTAL SETUP

We evaluate MixDiff within the setting where the class names of OOD samples and the OOD labels are unavailable at train time. This is a more challenging experimental setting compared to the environment where the OOD class names or its instances are known during the training phase. We follow the same setup as in Esmaeilpour et al. (2022), and evaluate our method on five OOD detection benchmark datasets: CIFAR10 (Krizhevsky et al., 2009), CIFAR100 (Krizhevsky et al., 2009), CIFAR+10 (Miller et al., 2021), CIFAR+50 (Miller et al., 2021), TinyImageNet (Le & Yang, 2015).

Each dataset's ID and OOD (known and unknown) class splits are composed of as follows. **CIFAR10** (Krizhevsky et al., 2009): the dataset's 10 classes are randomly split into 6 ID classes and 4 OOD classes. **CIFAR100** (Krizhevsky et al., 2009): consecutive 20 classes are assigned to be ID classes and the remaining 80 classes are assigned to be OOD classes. **CIFAR+10** (Miller et al., 2021): 4 non-animal classes of CIFAR10 are ID classes, 10 randomly sampled animal classes from CIFAR100 are OOD classes. **CIFAR+50** (Miller et al., 2021): 4 non-animal classes of CIFAR10 are ID classes, 50 randomly sampled animal classes from CIFAR100 are OOD classes. **TinyImageNet** (Le & Yang, 2015): considers 20 randomly sampled classes as ID classes and the remaining 180 classes as OOD classes.

For CIFAR10, CIFAR+10, CIFAR+50 and TinyImageNet, we follow the same ID, OOD class splits as in Miller et al. (2021); Esmaeilpour et al. (2022). For CIFAR100, we use the same class splits as in Esmaeilpour et al. (2022). Each dataset contains 5 splits, except for CIFAR+50, which is consisted of only one ID, OOD class split. Figure 9a shows each method's average AUROC scores averaged over the five datasets. All of the results in the upper block of the Table 1 are from Esmaeilpour et al. (2022) except for ZOC and MixDiff+ZOC.

### E.2    EVALUATION METRICS

We compare our method with the baseline methods using the metrics that are commonly employed for OOD detection tasks. **AUROC** is Area Under the Receiver Operating Characteristic where the

receiver operating characteristic represents the relationship between false positive rate (FPR) and true positive rate (TPR) for all of the threshold range. **FPR95** denotes the False Positive Rate when the threshold satisfies 95% TPR. **AUCPR** is Area Under the Curve of Precision and Recall. It is a useful performance measure, especially with an imbalanced dataset. For AUCPR, we set the detection threshold to be the value that satisfies 95% TPR.

### E.3 HYPERPARAMETER SEARCH ON CALTECH101

We construct each known-unknown class split for Caltech101 dataset (Fei-Fei et al., 2004) by randomly sampling 20 classes as ID, and setting aside the rest as OOD, making a total of 3 splits. We conduct grid search over the following hyperparameter configurations: $M \in \{15, 10\}$, $N \in \{14, 9\}$, $R \in \{7, 5\}$, $\gamma \in \{2.0, 1.0, 0.5\}$. We use the numbers that evenly divide the interval $[0, 1]$ into $R + 1$ segments as the values of the Mixup ratios. For example, when $R = |\Lambda| = 3$, the set of Mixup ratios is $\Lambda = \{0.25, 0.5, 0.75\}$. We select the configuration with the highest average AUROC score for each method. For the environment where the model outputs are the logits, the resulting hyperparameters are $M = 15$, $N = 14$, $R = 7$, and $\gamma = 2$ for all methods.

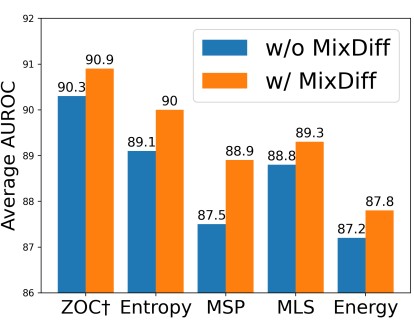

Figure 5: AUROC scores averaged over the five datasets.

For MSP+Entropy linear combination baseline, we tune the scaling factor $\eta = b \times 10^a$, by conducting grid search over the following configurations: $a \in \{-4, -3, -2, -1, 0, 1, 2, 3, 4\}$, $b \in \{1, 2, 3, 4, 5, 6, 7, 8, 9\}$ and the score to which $\eta$ is multiplied (MSP or Entropy).

### E.4 ADAPTATION OF MIXDIFF WITH ZOC

ZOC (Esmaeilpour et al., 2022) utilizes a candidate OOD class name generator. MixDiff framework is applied to ZOC by averaging out each of the perturbed images' candidate OOD logits as follows: $\log(\frac{1}{C} \sum_{i=1}^{C} \exp(o_i))$ where $C$ and $o_i$ are the number of generated OOD class names from the image and the $i$-th OOD class logit, respectively. This effectively means that the logits in the perturbed oracle and target samples' outputs have a dimension of $K + 1$ instead of $K$ in the following equations: $O^*_{mir} = f(x^*_{mir}) \in \mathbb{R}^K$ and $O_{ir} = f(x_{ir}) \in \mathbb{R}^K$.

We 200 randomly chosen samples per split were used as ZOC's token generation module requires a large amount of computation to process the entire set of mixed images. Also, in this case, the hyperparameters were tuned on each of the target datasets to alleviate variability issues.

### E.5 PRACTICAL IMPLEMENTATION

For each target sample $x_t$, MixDiff generates $N$ mixed samples, each of which with $R$ Mixup ratios. Similarly, it generates $N \times R$ mixed samples for each of $M$ oracle samples. If we follow the in-batch setup where the samples that are in the same batch as the target sample are used as the auxiliary samples, MixDiff requires processing of $BNR + BMNR$ mixed samples, denoting the batch size as $B = N + 1$.

We avoid $BNR + BMNR$ repeated forward passes by putting each set of the entire Mixup results, including the ones that are mixed with itself, into two tensors of sizes that are prefixed with $(B, B, R)$ and $(B, M, B, R)$, one for the mixed images of targets and auxiliary samples, the other for the mixed images of oracles and auxiliary samples, respectively. After computing the yet-to-be-averaged MixDiff scores within a tensor of size $(B, B, R)$, we zero out the diagonal entries in the first two dimensions, $(B, B)$, eliminating the scores from the target images that are mixed with itself. Then, we take the average of the last two dimensions, $(B, R)$, yielding $B$ MixDiff scores for each of the $B$ test samples.

We also note that in practice the set of $\bar{O}^*_{ir}$ prediction scores corresponding to the oracle samples mixed with the other in-batch samples do not need to be computed for every single test batch. One can use a fixed set of samples as an auxiliary set and precompute each of the mixed oracle logits $\bar{O}^*_{ir}$

by mixing these samples with the oracle samples. When a test batch arrives, each of the samples in the batch will then be independently mixed with these fixed auxiliary samples. Not only does it reduce the compute cost, there is no dependency on the test batch size in regard to OOD detection performance, since the auxiliary samples are no longer drawn from the test batch.

### E.6   EXPERIMENTAL DETAILS ON ADVERSARIAL ATTACK

We take the same experimental setup as the OOD detection experiments with identical datasets and backbone model. We use projected gradient descent (PGD) attack (Madry et al., 2018) with $\epsilon = \frac{1}{255}$ and attack step size of 10. Following Chen et al. (2022a), cross entropy with the uniform distribution is used as the loss function when attacking ID samples, and Shannone entropy is used as the loss function when attacking OOD samples.

### E.7   EXPERIMENTAL DETAILS ON OUT-OF-SCOPE TASK

We run out-of-scope detection experiments using 4 intent classification datasets. CLINC150 (Larson et al., 2019) dataset is consisted of samples spanning across 10 general domains including "utility" and "travel", with each sample belonging to one of 150 intent classes. Banking77 (Casanueva et al., 2020) is a dataset specializing in banking domain and has 77 intent classes. ACID (Acharya & Fung, 2020) is an intent detection dataset with 175 intents, consisted with samples of customers contacting an insurance company. TOP (Gupta et al., 2018) is a dataset with the intents organized in a hierarchical structure and is consisted of queries related to navigation and event. For TOP dataset, we use the root node's intent as the intent label for the query, as in Yilmaz & Toraman (2022).

For CLINC150 and TOP datasets, we keep the original OOS intents in the OOS split. More specifically, CLINC150 dataset's "oos" class and TOP dataset's intent classes that are prefixed with "IN:UNSUPPORTED" (Yilmaz & Toraman, 2022). We also set aside 4 intents in TOP dataset that have too small number of samples to be reliably split into train and validation sets, as OOS intents. These intents are "IN:GET_EVENT_ATTENDEE", "IN:UNINTELLIGIBLE", "IN:GET_EVENT_ORGANIZER", and "IN:GET_EVENT_ATTENDEE_AMOUNT". This leaves the dataset with 12 original in-scope intent classes, excluding the OOS intent classes.

To assume an environment where the test time in-scope ratio is unknown, we evaluate OOS detection performance on multiple inner in-scope ratios, 25%, 50%, 75%, for each inner split. We leave out the splits with the number of inner in-scope intents less than 2. An intent classification model is trained for each of these inner in-scope splits following the same procedure as described above. After training, we perform OOS detection on the outer in-scope validation set and select the hyperparameter set with the highest average AUROC score.

We further split the train set of the in-scope samples into more in-scope, OOS splits and use these to search MixDiff's hyperparameters. For a given oracle sample, we use the other oracle samples in the same class as the auxiliary samples. For ease of comparison between the logit-based and probability-based OOD scoring functions, we take the setup where the model $f(\cdot)$'s outputs are in the logit space for both cases.

We explore three configurations with respect to the position of the auxiliary sample in a concatenated text pair: (1) prepending the auxiliary sample at the front of an oracle or the target sample; (2) appending the auxiliary sample at the end of an oracle or the target sample; (3) a combination of both, analogous to the setting of 2 Mixup ratios in image Mixup ($R = 2$). We conduct grid search over the following hyperparameters: $M \in \{5, 10, 15, 20, 25, 30\}$, $\gamma \in \{0.5, 1.0, 2.0\}$, and three auxiliary sample concatenation methods as described above. We note that the number of auxiliary samples is determined as $N = M - 1$, since we use the other oracle samples in the same class as the auxiliary samples. We provide the average AUROC scores for each in-score ratio in Table 6.

### E.8   VERIFICATION EXPERIMENT OF THE MOTIVATION

Our primary hypothesis is that OOD samples are more susceptible to perturbations compared to ID samples of the predicted class of the target (oracle). To test this hypothesis, we utilize class activation map (CAM) Chen et al. (2022b) to observe changes in the model's attention areas before

| Method | In-scope ratio | CLINC150 | Banking77 | ACID | TOP | Average |
|---|---|---|---|---|---|---|
| MSP (Hendrycks & Gimpel, 2017) | 25% | $93.07_{\pm1.8}$ | $84.29_{\pm3.7}$ | $89.39_{\pm1.6}$ | $93.68_{\pm4.5}$ | 90.11 |
| | 50% | $93.26_{\pm0.6}$ | $85.80_{\pm3.2}$ | $88.61_{\pm1.3}$ | $88.90_{\pm5.2}$ | 89.14 |
| | 75% | $92.74_{\pm0.8}$ | $86.20_{\pm3.6}$ | $88.93_{\pm1.8}$ | $87.44_{\pm7.0}$ | 88.83 |
| | Avg. | 93.02 | 85.43 | 88.98 | 90.01 | 89.36 |
| MLS (Hendrycks et al., 2022) | 25% | $93.06_{\pm2.0}$ | $83.01_{\pm3.8}$ | $88.96_{\pm1.4}$ | $93.10_{\pm4.5}$ | 89.53 |
| | 50% | $93.77_{\pm0.6}$ | $85.63_{\pm3.2}$ | $88.77_{\pm1.0}$ | $88.30_{\pm6.2}$ | 89.12 |
| | 75% | $93.85_{\pm0.8}$ | $86.43_{\pm3.8}$ | $89.00_{\pm1.5}$ | $88.77_{\pm6.1}$ | 89.51 |
| | Avg. | 93.56 | 85.02 | 88.91 | 90.06 | 89.39 |
| Energy (Liu et al., 2020) | 25% | $93.09_{\pm2.1}$ | $82.96_{\pm3.8}$ | $88.87_{\pm1.4}$ | $93.10_{\pm4.5}$ | 89.51 |
| | 50% | $93.82_{\pm0.6}$ | $85.64_{\pm3.2}$ | $88.70_{\pm1.0}$ | $88.30_{\pm6.2}$ | 89.12 |
| | 75% | $93.91_{\pm0.8}$ | $86.36_{\pm3.8}$ | $88.93_{\pm1.5}$ | $88.78_{\pm6.1}$ | 89.50 |
| | Avg. | 93.61 | 84.99 | 88.83 | 90.06 | 89.37 |
| Entropy (Lakshminarayanan et al., 2017) | 25% | $93.23_{\pm1.8}$ | $84.28_{\pm3.8}$ | $89.27_{\pm1.6}$ | $93.68_{\pm4.5}$ | 90.12 |
| | 50% | $93.52_{\pm0.6}$ | $86.02_{\pm3.3}$ | $88.53_{\pm1.2}$ | $88.91_{\pm5.2}$ | 89.25 |
| | 75% | $93.11_{\pm0.8}$ | $86.48_{\pm3.8}$ | $88.81_{\pm1.7}$ | $87.46_{\pm7.0}$ | 88.97 |
| | Avg. | 93.29 | 85.59 | 88.87 | 90.02 | 89.44 |
| MixDiff+MSP | 25% | $93.57_{\pm1.7}$ | $84.77_{\pm3.6}$ | $89.66_{\pm1.6}$ | $93.68_{\pm4.6}$ | 90.42 |
| | 50% | $93.57_{\pm0.6}$ | $86.11_{\pm3.0}$ | $88.77_{\pm1.2}$ | $89.65_{\pm4.6}$ | 89.53 |
| | 75% | $93.12_{\pm0.8}$ | $86.36_{\pm3.4}$ | $89.10_{\pm1.6}$ | $88.71_{\pm6.1}$ | 89.32 |
| | Avg. | 93.42 | 85.75 | 89.18 | **90.68** | 89.76 |
| MixDiff+MLS | 25% | $93.57_{\pm2.0}$ | $83.56_{\pm3.7}$ | $89.37_{\pm1.4}$ | $93.16_{\pm4.4}$ | 89.92 |
| | 50% | $94.02_{\pm0.6}$ | $86.02_{\pm3.3}$ | $89.01_{\pm1.0}$ | $88.84_{\pm5.8}$ | 89.47 |
| | 75% | $94.04_{\pm0.7}$ | $86.81_{\pm3.6}$ | $89.33_{\pm1.3}$ | $89.04_{\pm6.0}$ | **89.81** |
| | Avg. | 93.88 | 85.46 | **89.24** | 90.35 | 89.73 |
| MixDiff+Energy | 25% | $93.59_{\pm2.0}$ | $83.51_{\pm3.7}$ | $89.28_{\pm1.4}$ | $93.17_{\pm4.4}$ | 89.89 |
| | 50% | $94.01_{\pm0.6}$ | $86.27_{\pm2.9}$ | $88.95_{\pm1.0}$ | $88.83_{\pm5.8}$ | 89.52 |
| | 75% | $94.07_{\pm0.8}$ | $86.74_{\pm3.7}$ | $89.32_{\pm1.3}$ | $89.05_{\pm6.0}$ | 89.80 |
| | Avg. | **93.89** | 85.51 | 89.18 | 90.35 | 89.73 |
| MixDiff+Entropy | 25% | $93.70_{\pm1.7}$ | $84.79_{\pm3.7}$ | $89.55_{\pm1.6}$ | $93.70_{\pm4.5}$ | **90.44** |
| | 50% | $93.84_{\pm0.6}$ | $86.42_{\pm3.1}$ | $88.74_{\pm1.2}$ | $89.65_{\pm4.7}$ | **89.66** |
| | 75% | $93.48_{\pm0.8}$ | $86.74_{\pm3.6}$ | $89.09_{\pm1.5}$ | $88.68_{\pm6.2}$ | 89.50 |
| | Avg. | 93.67 | **85.98** | 89.13 | **90.68** | **89.87** |

Table 6: Average AUROC scores for out-of-scope detection task. The numbers on the right side of $\pm$ represent standard deviation. The numbers in the "Average" column are the average AUROC scores reported in that row. The numbers in a "Avg." row are the average of the AUROC scores reported in that column. The highest and second highest average AUROC scores are highlighted with **bold**, and underline, respectively.

and after the Mixup operation when inferring classes. The results of our experiment, which support our hypothesis, are presented in Figure 6.

The experiment was conducted using CLIP ViT-B/32 and followed the same settings as in the OOD detection experiments on CIFAR100. We initially collected OOD samples that were misclassified as ID by the MSP score, with the purpose of collecting samples for which the model exhibited high confidence. We then filtered these samples to include only those classes with at least five samples per class. For each sample, an auxiliary sample for mixup was randomly selected from an ID class, excluding the class with the highest confidence for that sample.

We compared the CAM of OOD samples before and after Mixup. The CAMs were processed through min-max normalization, and values below 0.8 were clipped to be zero. We then measured the $L_1$ distance between the two CAMs. To observe the difference in CAM before and after Mixup for OOD class samples, we used the text prompt of the class predicted by the model. Conversely, for ID class samples, we used the text prompt of the ground truth.

Figure 6 compares the average distance in CAM for each class, considering the prompts as either a high confidence class or a ground truth class. A smaller distance implies less variation due to perturbation, suggesting that the features that the model focuses on are highly relevant to the respective class. On the other hand, a larger distance indicates a greater variation due to perturbation, which could mean that the features that the model focuses on are either less relevant to the class or incorrectly identified as relevant features. The results in Figure 6 indicate how perturbations can be used

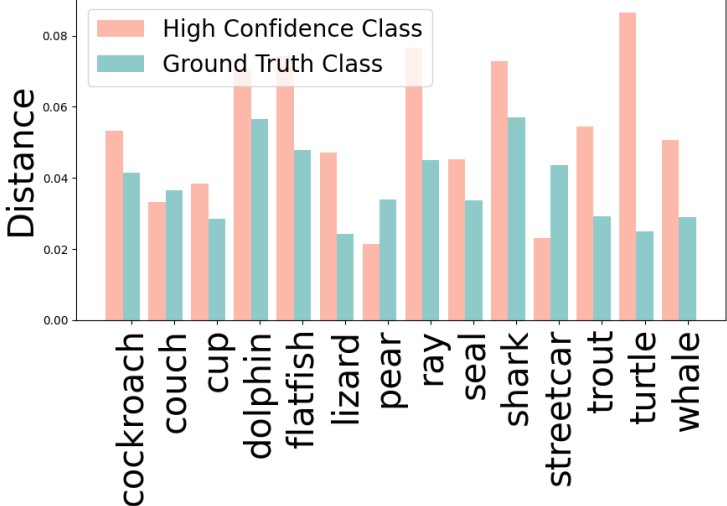

Figure 6: The average pixel-wise difference of the CAM images is measured before and after mixing the OOD samples with an auxiliary sample. The high confidence class and the ground truth class represent the classes used for the prompts in CAM. We observed how much fluctuations in the areas where the model focuses when the OOD samples are perturbed by arbitrary signals such as mixing with an auxiliary sample.

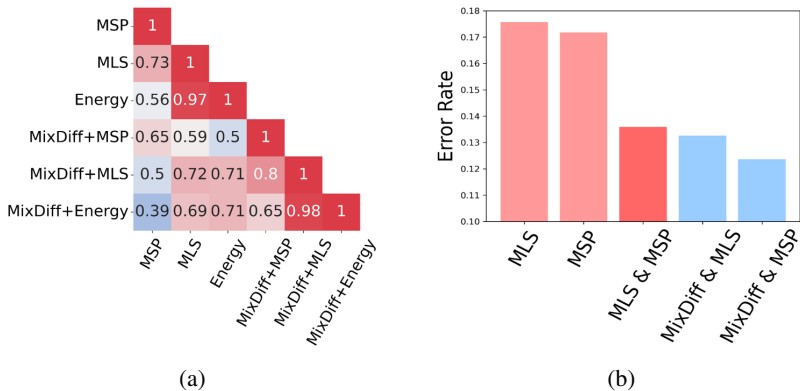

Figure 7: **(a)** Pearson correlation between the scores of different OOD scoring functions. **(b)** Error rate at TPR95 for each method. For multiple methods, error means both were incorrect.

to assess the reliability of the features that lead to a high level of confidence in the input predictions of the model.

### E.9 COMPARISON WITH OTHER OOD SCORING FUNCTIONS

To validate whether MixDiff scores have extra information which is not captured by existing other OOD scores, we calculate pair-wise correlation among OOD scores in Figure 8a, and evaluate the error rate of OOD detection by each OOD score in Figure 8b. All results from this subsection are derived from CIFAR100 test set. As shown in Figure 8a, MixDiff scores exhibit a weaker correlation with other OOD scores, which implies that MixDiff scores contain additional information that is absent in other scores. Consequently, MixDiff can correct certain wrong decisions of existing methods (verified in Figure 8b), when adopted with them together. From these results, we argue that the perturb-and-compare approach is helpful for stable OOD detection and MixDiff effectively provides such an advantage.

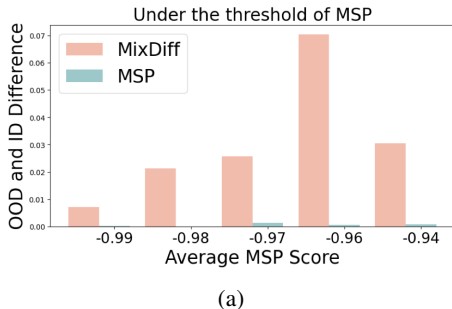 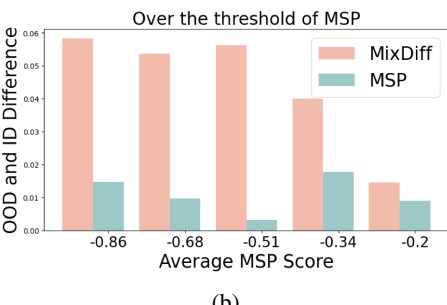

(a)                                                              (b)

Figure 8: Difference between the average uncertainty scores of OOD and ID samples belonging to a given interval of MSP score. The $x$-axis represents the average MSP score of the interval. **(a)** Under the threshold with the threshold set by TPR95 of MSP. **(a)** Over the threshold with the threshold set by TPR95 of MSP.

We divide the MSP score into five intervals of the same length and plot the difference of the average of the scores of OOD and ID samples in the same interval. We also plot the difference of the average of the MixDiff scores of OOD and ID samples belonging to the same MSP score interval. Figure 8 shows that for similar values MSP score, the uncertainty score from MixDiff among the OOD samples is significantly higher than that of the ID samples. This demonstrates that, even when two ID, OOD samples' MSP scores are almost identical, the MixDiff scores can still provide a discriminative edge.

## E.10 SENSITIVITY ANALYSIS

Figure 9b shows changes in AUROC score on the CIFAR100 dataset in regard to the number of Mixup ratios, $R$. We fix the other hyperparameters and only vary $R$. For all OOD scoring functions, logits are used as the model $f(\cdot)$'s outputs when computing perturbed oracles' OOD scores.

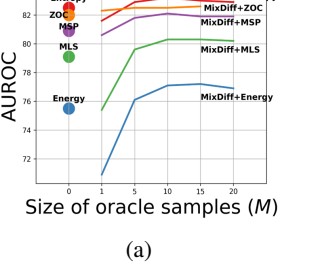 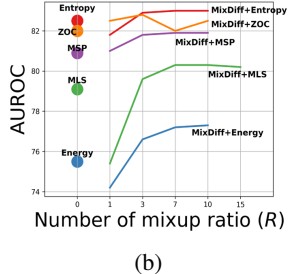

(a)                                         (b)

## E.11 COMUTATIONAL COST ANALYSIS

Figure 10 shows AUROC scores of MixDiff+Entropy for various values $R$ and $N$ evaluated on CIFAR100. MixDiff starts to outperform the entropy score with only two additional forward passes ($N = 2$, $R = 3$).

Figure 9: **(a)** Performance change in regard to the number of oracle samples, $M$. **(b)** Performance change in regard to the number of Mixup ratios, $R$.

The model outputs from $f(\cdot)$ are prediction probabilities and the number of oracle samples, $M$, is fixed at 15.

## E.12 PROCESSING TIME ANALYSIS

We analyze the average processing time required to process one target sample. We fix the number of oracle samples, $M$, to be 15 and use the other oracle samples as the auxiliary samples ($N$=14). This is the same as the oracle as auxiliary setup in the ablation studies portion of the main paper (Section 3. Figure 11 depicts the average processing time along with the number of Mixup ratios, $R$. As the additional perturbed samples can be effectively processed in parallel, MixDiff's effectiveness can be exploited without incurring prohibitive processing time as the rapid increase in performance at the small values of $R$ indicates. When we allow multiple target samples to be batched together, MixDiff's processing time further decreases (MixDiff BS=100).

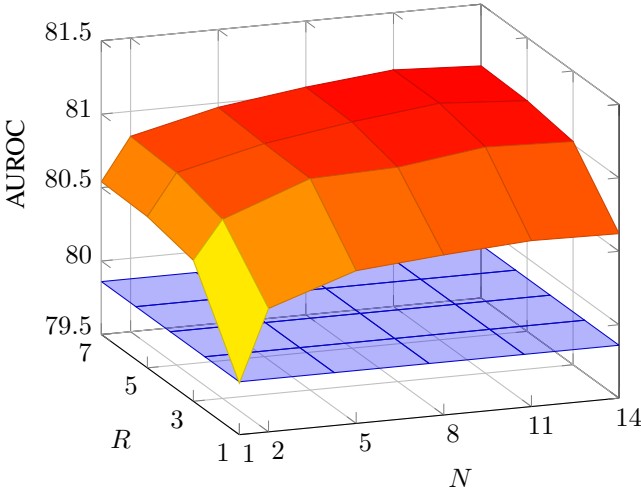

Figure 10: AUROC scores of MixDiff+Entropy with varing values of $N$ and $R$ (top). AUROC score of Entropy (bottom). Both methods are evaluated on CIFAR100.

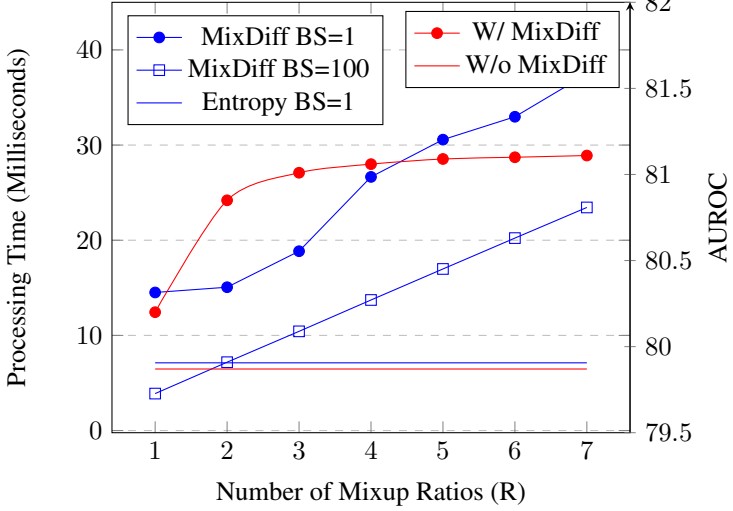

Figure 11: Blue lines represent the average processing time per target sample. BS denotes the batch size of target samples. Red lines represent AUROC scores of MixDiff+Entropy and entropy OOD scoring function evaluated on CIFAR100.

# F  QUALITATIVE ANALYSIS

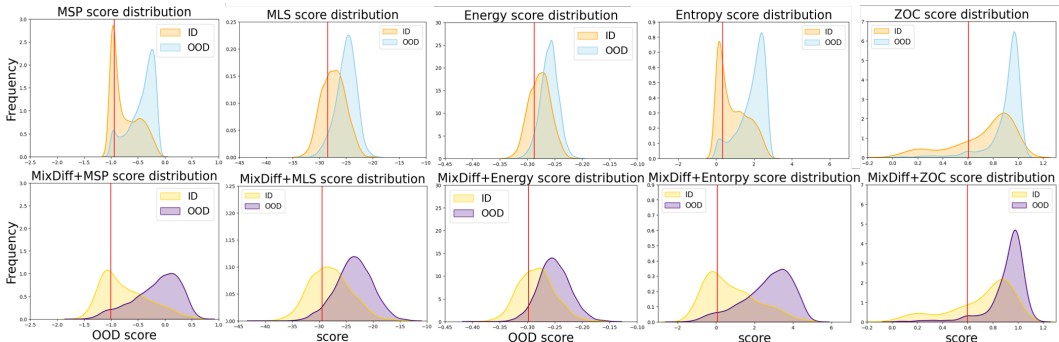

Figure 12: Visualize the distribution of the OOD score using a kernel density estimate plot. The density curves in each column denote the distribution of the base OOD score functions (MSP and Entropy), MixDiff scores, and MixDiff + base scores, respectively. The red vertical line in the figures above represents the 95% TPR threshold for each OOD score (base OOD, MixDiff, and MixDiff+base OOD) function.

|  | MSP | MixDiff+MSP | MLS | MixDiff+MLS | Energy | MixDiff+Energy | Entropy | MixDiff+Entropy | ZOC | MixDiff+ZOC |
|---|---|---|---|---|---|---|---|---|---|---|
| Threshold (95 % TPR) | -0.938 | -1.007 | -28.44 | -29.42 | -0.287 | -0.297 | 0.358 | 0.071 | 0.604 | 0.598 |
| ID over threshold (↓) | 0.688 | **0.663** | 0.667 | **0.625** | 0.684 | **0.645** | 0.656 | **0.628** | 0.731 | **0.725** |
| ID under threshold (↑) | 0.296 | **0.322** | 0.322 | **0.360** | 0.304 | **0.340** | 0.331 | **0.358** | 0.263 | **0.268** |
| OOD over threshold | 0.949 | 0.950 | 0.947 | 0.947 | 0.945 | 0.947 | 0.938 | 0.947 | 0.947 | 0.949 |
| OOD under threshold | 0.048 | 0.047 | 0.050 | 0.049 | 0.051 | 0.048 | 0.060 | 0.050 | 0.051 | 0.049 |

Table 7: The integral of the density curve (Figure 12) of the ID and OOD samples divided by the threshold for each methodology. (↓ means lower is better and ↑ means higher is better.)

## F.1  THE INTEGRAL OF THE DENSITY CURVE

Figure 12 plots the score distribution of the base OOD score function and MixDiff+base function. Table 7 shows the area under the distribution curves of In-distribution (ID) and Out-of-distribution (OOD) samples separated by the threshold for each approach. The distribution in Figure 12 shows that the base functions generally have a narrow range of scores, while MixDiff + base functions have a relatively wide range. We interpret this result as the MixDiff score being added to the base score, which adjusts the score of ID and OOD to alleviate overlapping. In Table 7, adding the MixDiff score increases the distribution area of OOD samples with higher scores and ID samples with lower scores relative to the threshold while decreasing the distribution area of OOD samples with lower scores and ID samples with higher scores. For all OOD scoring functions, logits are used as the model $f(\cdot)$'s outputs when computing perturbed oracles' OOD scores.

## F.2  3D LOGITS AND AUXILIARY SAMPLES

To see the effect of MixDiff in the logit level, we plot the logits of the target, oracle, and the corresponding mixed objects in Figure 13. For all OOD scoring functions, logits are used as the model $f(\cdot)$'s outputs when computing perturbed oracles' OOD scores.

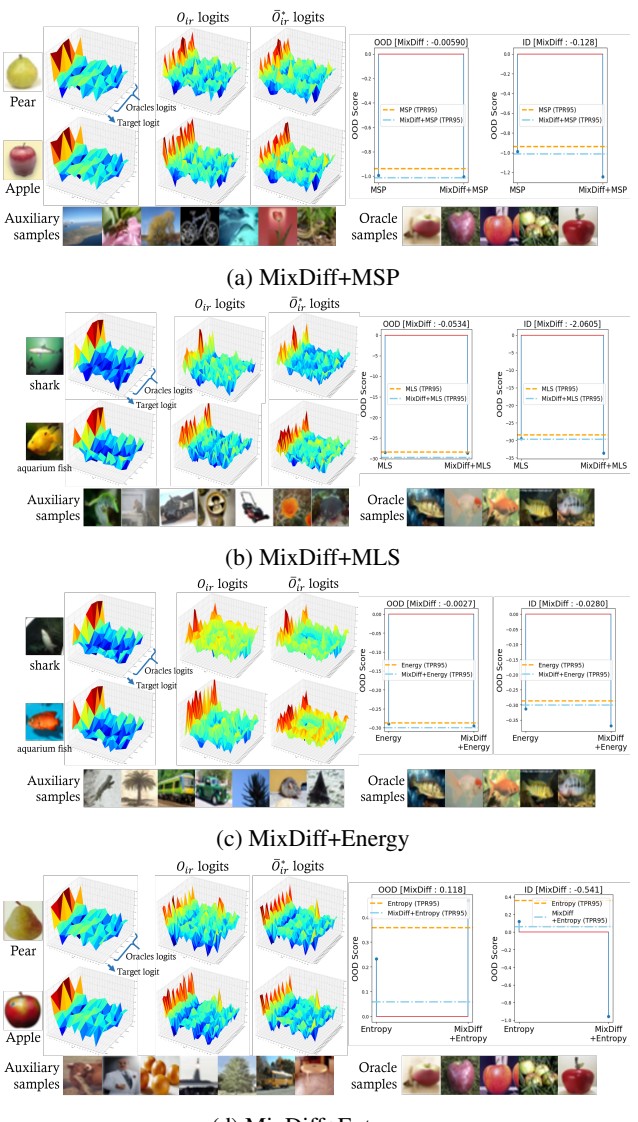

Figure 13: Logit level changes when mixing the same auxiliary samples for target and oracle. The first row of 3d logits in Figure 13a-13c implies even though there is an OOD sample that is indistinguishable from the oracles at the logit level, the difference could be captured by mixing up with auxiliary samples. The figures in the second row in Figure 13a-13c show the 3d logits of the ID sample whose class is the same as the oracle samples. The two graphs to the right of each figure show the OOD scores and thresholds for the base OOD score function and MixDiff+base for the OOD and ID target samples, respectively.

## G    LIMITATIONS AND FUTURE WORK

**Time and space complexity**    MixDiff is effective at bypassing a black-box model's access restriction for OOD detection, but bypassing the access restriction comes with a certain computational overhead. For each target sample $x_t$, MixDiff requires processing of $N \times R$ mixed samples. While these samples can be effectively processed in parallel and the MixDiff framework outperforming the baselines only with small values of $R$, it nonetheless remains as a drawback of the MixDiff framework. Further research is called for reducing the computational and space complexity of MixDiff framework. We note that the mixed oracle, auxiliary samples need only be processed once and require negligible compute and memory thereafter.

**Intermediate state extraction**    MixDiff framework as we have presented extracts the model's prediction scores of oracle-auxiliary mixed samples as intermediate states. These intermediate states are then averaged along the number of oracle samples to reduce the variance induced by instance specific characteristics of the oracle samples. The intermediate state extraction procedure can be easily generalized to incorporate other OOD score functions. Taking a concrete example, in the case of Mahalanobis distance based OOD detection methods (Lee et al., 2018; Chen et al., 2022c; Ren et al., 2021), the intermediate states could be the features that are used to construct the class-conditional Gaussian distributions. One extreme case would be taking the final OOD scores of oracle-auxiliary mixed samples as the intermediate states and averaging these OOD scores along the number of oracle samples. We have run extensive experiments on the output-based OOD scores, such as MSP and MLS, and leave the intermediate state extraction choices in other OOD score functions as future work.

**Other forms of inputs**    MixDiff framework can be easily extended to incorporate inputs from other modalities. The experiments on the out-of-scope detection task serve as an example of these kinds of extensions. This input level Mixup makes the framework applicable to environments where the access to the model parameters cannot be assumed. It also grants the freedom to design better mixup methods that are specific to the format of the input or the task at hand. But this freedom comes at the cost of having to devise a mixup mechanism for each input format and task. For example, the simple concatenation of samples that we have utilized in OOS detection experiments on the out-of-scope detection task has the limitation that it cannot be applied if the input sequence is too long, due to quadratic time and space complexity of Transformers (Vaswani et al., 2017). A future research can be conducted in the direction of forgoing this self-imposed requirement that the mixup operation be performed in the input level. Allowing mixup of model activations may pave the way for more broadly applicable form of the framework, albeit with a possible degradation in performance.

**Selection of auxiliary samples**    We have experimented in the paper with three auxiliary sample selection methods, one using the in-batch samples and the other two using the oracle or random ID samples as the auxiliary samples. Our preliminary experiments showed performance degradation when the number of auxiliary samples, $N$, is too small. We hypothesize that this is due to the fact that while on average MixDiff can effectively discern the overemphasized features by comparing mixed oracle-auxiliary and mixed target-auxiliary samples, there is a certain degree of variance in the MixDiff score, requiring $N$ and, to some degree, $R$ to be over certain value for reliable performance. There may be an auxiliary sample that is more effective at discerning an overemphasized feature of a given target sample, but this is subject to change depending on the target sample. We leave the exploration of better auxiliary sample selection methods, either by careful selection of auxiliary samples or by making the procedure instance-aware and possibly learnable, as future work.

