# OpenReview forum: "Perturb-and-Compare Approach for Detecting Out-of-Distribution Samples in Constrained Access Environments"
_ICLR.cc/2024/Conference — Submitted to ICLR 2024_

### Official Review · Reviewer_EmiV · 2023-10-22

**Soundness:** 2 fair
**Presentation:** 2 fair
**Contribution:** 2 fair
**Rating:** 3
**Confidence:** 4

**Summary:**

This paper proposes a perturbation-based OOD detection method to detect potential OOD samples when the model parameters cannot be accessed.

**Strengths:**

The tasks and scenarios proposed in the paper seem to have more application value at present.

**Weaknesses:**

1. The core motivation of the paper is that perturbed OOD samples will reduce more confidence. But this motivation is not intuitive and cannot be verified. This motivation fails to show its competitiveness compared to maximum class probabilities or predicted entropy.
2. Compared with directly using ood detection methods such as maximum class probability or predicted entropy, the complexity of the proposed method is very high, but according to the experimental results, the performance improvement is very limited.
3. When only the predicted labels of the model can be obtained, the performance of the model is very limited. And when the predicted probability of the model is obtained, it is no different from the previous OOD method without training, and I cannot see its significant advantage.

**Questions:**

None

---

> ### Author Response · Authors · 2023-11-23
> **Response to Reviewer EmiV [1/3]**
>
> We are pleased to know that you found OOD detection problem in constrained access scenarios valuable. We found the comments that you have provided helpful and appreciate your valuable time spent on improving our work. We have made following revisions to our manuscript based on the suggestions in the review that you have kindly provided to us.
>
> > 1. The core motivation of the paper is that perturbed OOD samples will reduce more confidence. But this motivation is not intuitive and cannot be verified. This motivation fails to show its competitiveness compared to maximum class probabilities or predicted entropy.
> >
>
> Thank you for suggesting the direction we can improve our paper. The paper’s main motivation is that the contributing features that induce misclassification are less robust to perturbations. Misclassification here refers to misclassification as ID or OOD. We revised Figure 1 to include an actual example of this scenario [(Section 1, Figure 1a, page 2)]. To visually illustrate the locations of the model’s focus in the images, we utilized class activation maps. Figure 1a depicts that a picture of a train had a high confidence score because it resembled an ID class bus, despite it being an OOD sample. The features that led to the high confidence score for the bus are less robust to perturbations compared to the features in an actual picture of a bus under the same perturbation, as seen by the class activation maps and the model’s prediction scores.
>
> We have also included a verification experiment in [(Section 1, Figure 1b, page 2), (Appendix E.8, Figure 6, pages 21-23)] to provide evidence for the intuition that contributing features in a misclassified sample is more sensitive to perturbations. In the experiment, we demonstrate that the locations in a sample on which the model has focused are more sensitive to perturbations when these features do not belong to the ground truth class. More specifically, for a given OOD sample with a high confidence score, we calculated the distance of the class activation maps of the predicted class before and after a perturbation. Compared to the class activation maps of the OOD sample’s actual ground truth class, these predicted class’s class activation maps tended to be more brittle to perturbations, despite having an initial high confidence prediction score for that class. This suggests that even though certain features in a misclassified OOD sample had a strong impact on the model’s high degree of confidence, these features have a tendency to be less robust to perturbations.
>
> We have conducted another experiment to further clarify this aspect and revised the manuscript to incorporate these findings in [(Section 4.5, Table 4, pages 8-9), (Appendix E.6, page 21)]. The experimental setup is based on adversarial attack scenarios. In an adversarial attack, attackers aim to fool the OOD detector, so that an ID sample is detected as OOD, and vice versa. This is commonly done by adding a small modification to a target sample that is indistinguishable to the human eye. It can be viewed that these modifications inject artificial features in the target sample so as to induce overconfident or underconfident outputs from the model. The experimental results demonstrate  MixDiff’s superiority on robustness to adversarial attacks. This indicates that the contributing features that induce ID/OOD misclassification are less robust to perturbations and that MixDiff can effectively exploit such brittleness. The AUROC scores are presented in the table below.
>
> | Method | CIFAR10 |  |  |  | CIFAR100 |  |  |  |
> | --- | --- | --- | --- | --- | --- | --- | --- | --- |
> |  | Clean | In | Out | Both | Clean | In | Out | Both |
> | Entropy | 89.88 | 47.42 | 13.77 | 2.678 | 79.87 | 36.86 | 14.38 | 2.21 |
> | MixDiff+Entropy | 90.64 | 54.71 | 31.77 | 9.084 | 81.11 | 47.42 | 31.40 | 9.08 |
> | MixDiff Only | 88.16 | 61.00 | 40.28 | 20.45 | 78.05 | 58.84 | 44.19 | 27.48 |

---

> ### Author Response · Authors · 2023-11-23
> **Response to Reviewer EmiV [2/3]**
>
> > 2. Compared with directly using ood detection methods such as maximum class probability or predicted entropy, the complexity of the proposed method is very high, but according to the experimental results, the performance improvement is very limited.
> >
>
> We are grateful for the reviewer for raising an important question regarding the computational aspect of our approach. We have included an analysis on computational cost in [(Appendix E.11, Figure 10, pages 24-25)]. For a given target sample, MixDiff requires processing additional $N \cdot R$  samples, where $N$ is the number of auxiliary samples, and $R$ is the number of Mixup rates. A more detailed description can be found in Appendix E.6. To demonstrate that MixDiff can still be beneficial even with a small computational budget, we performed additional analysis on the effect of $N$  and $R$  on the OOD detection performance. The results show that MixDiff’s performance is on par with the base score with one additional forward pass, $(N=1, R=1)$, and starts to outperform it with only two additional forward passes, $(N=1, R=2)$. The computational cost can be adjusted to fit a given budget while still enjoying the benefits of enhanced detection performance of the base score.
>
> We have also revised the manuscript to include analysis on the average time required to process the additional samples in [(Appendix E.12, Figure 11, pages 24-25)]. As discussed in Section 3, these samples can be processed in parallel, as the order in which they are processed does not matter. We have run additional analysis on the processing time. First, we put all of the target sample’s perturbed samples in a single batch and measure the average processing time per target sample and AUROC score with varying values of $R$. At a lower range of $R$, the detection performance increases rapidly while the processing time is somewhat stagnant, suggesting that the performance improvement from MixDiff can be gained without incurring a prohibitive processing time.

---

> ### Author Response · Authors · 2023-11-23
> **Response to Reviewer EmiV [3/3]**
>
> > 3. When only the predicted labels of the model can be obtained, the performance of the model is very limited. And when the predicted probability of the model is obtained, it is no different from the previous OOD method without training, and I cannot see its significant advantage.
> >
>
> We appreciate the detailed feedback regarding the performance aspect of our work. As we discuss in the Sections 1 and 2, MixDiff was proposed with the aim of enhancing output-based OOD detection functions without any access to the models’ internal states. This includes the ability to discriminate a pair of OOD and ID samples where the two samples have an almost identical model outputs, since in this case, output-based scores would struggle to distinguish the OOD sample. The environment where only the predicted labels are provided serves as an extreme case where all of the model’s confidence scores are hidden from the user. The experiments on this environment effectively show how MixDiff deals with the restriction that the model’s confidence scores for all of the samples are identical. And even in this extreme case, where none of the baseline methods were directly applicable, MixDiff was able to gather additional information by perturbing the target sample and comparing the perturbed target sample’s confidence scores with perturbed oracle samples, while only accessing the input level samples. This was evidenced by MixDiff gaining more than 10 AUROC score compared to random guessing.
>
> As for the environment where only the prediction probabilities are accessible, MixDiff+MSP score improves the MSP score by 1.4 AUROC on average, while MixDiff+Entropy score improves the entropy score by 0.8 AUROC on average [(Section 4, Table 3, page 8)]. In addition to that, we have added another baseline, MCM [1], to better demonstrate the effectiveness of MixDiff’s OOD detection performance [(Section 4, Table 1, page 7)]. While MCM exhibits a strong performance, outperforming all of the other four output-based baselines, this method requires an access to the logit space, which may not be available from a black-box model. However, the entropy score equipped with MixDiff outperforms MCM on 4 out of 5 datasets as well as on average AUROC over the five datasets, while solely relying on the model’s prediction probabilities. This is in contrast to the case where a linear combination of MSP and the entropy scores not yielding any improvement on the entropy score [(Section 4, Table 3, page 8)]. These results suggest that MixDiff scores contain additional information which has been effectively gained from MixDiff’s perturb-and-compare approach. We present each method's the average AUROC scores in the table below. We have revised Section 4 of the manuscript to reflect the additional experiments.
>
> | Method | CIFAR10 | CIFAR100 | CIFAR+10 | CIFAR+50 | TinyImageNet | Average |
> | --- | --- | --- | --- | --- | --- | --- |
> | MCM (Ming et al., 2022) | 90.6±2.9 | 80.3±2.1 | 96.9±0.8 | 97.0 | 83.1±2.2 | 89.6 |
> | MixDiff+MCM | 91.4±1.8 | 81.4±2.6 | 97.5±0.9 | 97.7 | 83.9±2.2 | 90.4 |
> | Entropy | 89.9±2.6 | 79.9±2.5 | 96.8±0.8 | 96.8 | 82.2±2.3 | 89.1 |
> | MixDiff+Entropy (Pred. prob.) | 91.1±1.6 | 80.9±2.6 |  97.1±0.8 | 97.3 | 82.9±2.3 | 89.9 |
> | DML (Zhang & Xiang, 2023) | 87.8±3.0 | 80.0±3.1 | 96.1±0.8 | 96.0 | 84.0±1.2 | 88.8 |
> | ASH (Djurisic et al., 2023) | 85.2±3.8 | 75.4±4.4 | 92.5±0.9 | 92.4 | 77.2±3.1 | 84.5 |
>
> We perform an additional analysis [(Appendix E.9, Figure 8, pages 23-24)] to qualitatively showcase the contribution of MixDiff score to other base scores. To that end, we divide the MSP score’s values into certain number of consecutive intervals, so that the MSP values of a given interval would be similar to each other. For each interval, we calculate the mean of MSP scores of ID and OOD samples, respectively. The mean of MSP scores for ID and OOD samples are expected to be similar in value as these samples belong to the same interval. Next, for each interval of MSP, we calculate the mean of MixDiff scores of ID and OOD samples belonging to that interval. Finally, when we plot the difference of the mean of ID and mean of OOD samples for each interval, the MixDiff’s difference is significantly higher. This qualitatively demonstrates that even when two MSP scores are almost identical, MixDiff scores can provide a discriminative edge over the MSP score, while only modifying the input level sample.
>
> [1] Delving into out-of-distribution detection with vision-language representations. Advances in Neural Information Processing Systems, 35:35087–35102, 2022.

---

### Official Review · Reviewer_hNhc · 2023-10-28

**Soundness:** 3 good
**Presentation:** 3 good
**Contribution:** 2 fair
**Rating:** 5
**Confidence:** 4

**Summary:**

The paper proposes an OOD detection framework, MixDiff, that is applicable even when the model parameters or its activations are not accessible to the end user. To bypass the access restriction, MixDiff applies an identical input-level perturbation to a given target sample and an in-distribution (ID) sample that is similar to the target and compares the relative difference of the model outputs of these two samples. MixDiff is model-agnostic and compatible with existing output-based OOD detection methods. We provide theoretical analysis to illustrate MixDiff’s effectiveness at discerning OOD samples that induce overconfident outputs from the model and empirically show that MixDiff consistently improves the OOD detection performance on various datasets in vision and text domains

**Strengths:**

1. The paper is written well and is easy to understand.
2. The studied problem is very important.
3. The results seem to outperform state-of-the-art.

**Weaknesses:**

1. The motivation discussed in the introduction is somewhat opposite to the existing empirical findings on OOD detection with perturbed data. see [1], which suggests that the perturbed OOD data will also be predicted with overconfident probabilities for neural networks.
2. The compared baselines are not state-of-the-art. The authors are suggested to compare with more recent strong methods, such as those listed in [3,4]
3. The authors are suggested to justify the assumptions used in the theory in a formal way.
4. What is the computation cost of the proposed methods? It seems like the current method will require multiple forward/backward steps of the neural net during inference.

[1] Robust Out-of-distribution Detection for Neural Networks

[2] ATOM: Robustifying Out-of-distribution Detection Using Outlier Mining

[3] OpenOOD: Benchmarking Generalized Out-of-Distribution Detection

[4] OpenOOD v1.5: Enhanced Benchmark for Out-of-Distribution Detection

**Questions:**

see above

---

> ### Author Response · Authors · 2023-11-23
> **Response to Reviewer hNhc [1/2]**
>
> We appreciate the directions of improvements for our manuscript that you have suggested. It has helped strengthen our manuscript in various aspects. We are also glad that you think that improving OOD detection performance in black-box environment is very important and that our method advances the state-of-the-art in this important area. Below are the summary of the revisions we have made to our manuscript incorporating the helpful feedback in the review.
>
> > 1. The motivation discussed in the introduction is somewhat opposite to the existing empirical findings on OOD detection with perturbed data. see [1], which suggests that the perturbed OOD data will also be predicted with overconfident probabilities for neural networks.
> >
>
> Thank you for pointing out these fine points. In adversarial attack scenarios, the attacker tries to find a perturbation for a given OOD sample that induces confident prediction probability from the model. This is commonly achieved by following the model’s gradient that guides the perturbation to the direction that increases the model’s certainty regarding the sample. This can be seen as inserting an artificial feature that is optimized for this specific sample. You are correct in noting that a perturbation can increase the model’s confidence for a given target sample.
>
> However, in our case, the purpose of applying perturbations is not necessarily to increase or decrease the target sample’s confidence. It is to see how robust the features contained in the target sample are. This is achieved by applying undirected perturbations to the target sample and comparing how the same perturbations affect an actual ID sample that is deemed to possess the same features as determined by the predicted class label of the target sample. Our theoretical analysis suggests that for a given OOD sample with a high confidence score, it is not unreasonable to find an auxiliary sample that will reduce the confidence score of the target compared to an oracle when both are mixed with the same auxiliary sample.
>
> To reiterate, the perturbations in adversarial attack scenarios can be regarded as injecting artificial features to a target image, while our perturbations can be viewed as more of an undirected noise designed to test whether a certain feature is robust enough to be considered as real features. Inspired by this idea, we have conducted an additional experiment [(Section 4.5, Table 4, pages 8-9), (Appendix E.6, page 21)] designed to test MixDiff’s robustness to these artificial features and the experimental results indicate that MixDiff is significantly more effective at discerning these artificial features than a baseline score. We present the AUROC scores in the table below.
>
> | Method | CIFAR10 |  |  |  | CIFAR100 |  |  |  |
> | --- | --- | --- | --- | --- | --- | --- | --- | --- |
> |  | Clean | In | Out | Both | Clean | In | Out | Both |
> | Entropy | 89.88 | 47.42 | 13.77 | 2.678 | 79.87 | 36.86 | 14.38 | 2.21 |
> | MixDiff+Entropy | 90.64 | 54.71 | 31.77 | 9.084 | 81.11 | 47.42 | 31.40 | 9.08 |
> | MixDiff Only | 88.16 | 61.00 | 40.28 | 20.45 | 78.05 | 58.84 | 44.19 | 27.48 |
>
> > 2. The compared baselines are not state-of-the-art. The authors are suggested to compare with more recent strong methods, such as those listed in [3,4]
> >
>
> We have run additional experiments Section 4, Table 1, page 8)] with two recently proposed methods that on model’s inner activations, namely  DML [1] and ASH [2] and added the results in Table 3. For fair comparison with zero-shot OOD detection methods, we applied these methods to zero-shot CLIP classifier. However, these methods did not outperform the baseline scores. We suspect this is because they were validated with finetuned models such as ResNet50 finetuned on the ID dataset, whereas our experimental setup focuses on zero-shot OOD detection. This result is in line with previous findings [3] where a similar method, ReAct [4], did not outperform the Energy score in zero-shot OOD detection task.
>
> This has prompted us to run another experiment [(Section 4, Table 1, page 7)] to evaluate MCM [5]. MCM has shown strong performance on zero-shot OOD detection task. However, it requires an access to the logit space, which may not be accessible in a black-box API. This method outperforms the baseline scores that are based on the model’s prediction probabilities, namely, Entropy and MSP, as well as two other logit-based OOD scores (MLS, Energy). However, MixDiff equipped with entropy score outperforms MCM, on 4 out of 5 datasets as well as on average over the 5 datasets, while solely relying on model’s prediction probabilities. The results suggest that MixDiff’s perturb-and-compare approach is effective at gathering more discriminative information that are not available in prediction probabilities alone. Moreover, MixDiff further improves MCM’s detection performance when applied to it.

---

> ### Author Response · Authors · 2023-11-23
> **Response to Reviewer hNhc [2/2]**
>
> > 3. The authors are suggested to justify the assumptions used in the theory in a formal way.
> >
>
> First of all, we appreciate your detailed review and valuable feedback. In light of your review concerning the theorem, we acknowledge the need to reassess the underlying assumptions to ensure the validity of our theorem. Thanks to your feedback, revisiting the proof of our theorem allowed us to refine our approach to the proof of Theorem 1. This process enabled us to demonstrate that our proposed theorem can hold under mode general and reasonable conditions, leading to its significant advancement.
>
> In our original theorem, we’ve revisited the following assumptions
> $0 < f(x_m) < f(x_t), f(x_t)=f(x_m)+c$, where $c>0.5(1-\lambda)\sigma'(f(x_m))$.
>
> $c$ represents the extent to which the confidence of a target sample exceeds that of an arbitrary oracle sample, particularly when the target sample is a ‘hard OOD sample’. This refers to samples that would be mistakenly classified as in-distribution by base OOD score functions.
>
> In the revised version of our theorem [(Section 3.1, Theorem 1, pages 5-6), (Appendix C, Theorem 1, pages 17-18)], we have theoretically demonstrated that our method remains valid even under more generalized conditions.
>
> We continue to maintain the assumption that $f(x_t)$ is a hard OOD target sample. However, we have revised the condition for $c$ to simply require that it be greater than 0. Additionally assuming the confidence of any arbitrary oracle sample $f(x_m)$ is sufficiently large, such that $0<\frac{\sigma''(f_t)}{\sigma''(f_m)}<1$, we have verified the existence of an auxiliary sample $x_i$ such that
>
> $h(f(x_t))-h(f(x_m))+\sum_{l=1}^3(\omega_l(x_t,x_i)-\omega_l(x_m,x_i))>0$.
>
> Please refer to Appendix C for the revised proof of Theorem 1.
>
> > 4. What is the computation cost of the proposed methods? It seems like the current method will require multiple forward/backward steps of the neural net during inference.
> >
>
> Thank you for raising a question regarding this important aspect. As we discuss in Appendix G, MixDiff’s effectiveness at gaining more discriminative information from a black-box model comes with a certain computational overhead, as we probe the model by passing multiple perturbed samples to the black-box model. For a given target sample, MixDiff requires processing additional $N \cdot R$  samples, where $N$ is the number of auxiliary samples, and $R$ is the number of Mixup rates. A more detailed description can be found in Appendix E.5. We note that no backward step is performed as MixDiff does not require any gradient information, which makes it applicable to black-box environments.
>
> To demonstrate that MixDiff can still be beneficial even with a small computational budget, we performed an additional analysis [(Appendix E.11, Figure 10, pages 24-25)] on the effect of $N$  and $R$  on the OOD detection performance. The results show that MixDiff’s performance is on par with the base score with one additional forward pass, $(N=1, R=1)$, and starts to outperform it with only two additional forward passes, $(N=1, R=2)$. The computational cost can be adjusted to fit a given budget while still enjoying the benefits of enhanced detection performance of a base score. We have added the results of the analysis in Appendix E.11.
>
> Another important aspect is the time required to process the additional samples. We added a plot detailing the performance and processing time trade-off in [(Appendix E.12, Figure 11, pages 24-25)]. As discussed in Section 3, MixDiff’s perturbed samples can be processed in parallel, as the order in which they are processed does not matter. We have run additional analysis on the processing time. First, we put all of the target sample’s perturbed samples in a single batch and measure the average processing time per target sample and AUROC score with varying values of $R$. At a lower range of $R$, the detection performance increases rapidly while the processing time is somewhat stagnant, suggesting that the performance improvement from MixDiff can be gained without incurring a prohibitive processing time.
>
> [1] Decoupling maxlogit for out-of-distribution detection. In Proceedings of the IEEE/CVF Conference on Computer Vision and Pattern Recognition (CVPR), pp. 3388–3397, June 2023.
>
> [2] Extremely simple activation shaping for out-of-distribution detection. In The Eleventh International Conference on Learning Representations, 2023.
>
> [3] Clipn for zero-shot ood detection: Teaching clip to say no. In Proceedings of the IEEE/CVF International Conference on Computer Vision, pp. 1802–1812, 2023.
>
> [4] React: Out-of-distribution detection with rectified activations. In Advances in Neural Information Processing Systems, 2021.
>
> [5] Delving into out-of-distribution detection with vision-language representations. Advances in Neural Information Processing Systems, 35:35087–35102, 2022.

---

### Official Review · Reviewer_LGSq · 2023-10-30

**Soundness:** 3 good
**Presentation:** 4 excellent
**Contribution:** 3 good
**Rating:** 8
**Confidence:** 3

**Summary:**

This paper proposes a new OOD detection framework called MixDiff. MixDiff does not require knowledge of target models and only considers access to the input and output of an ML model. Specifically, MixDiff applies the same perturbation to a target sample and a trustworthy example and identifies an anomaly by comparing the relative difference between the model outputs. Empirically, the authors show that MixDiff performs well on various tasks.

**Strengths:**

(1) The paper is overall well-written and easy to read.

(2) The proposed method is well-motivated by Figure 1 and the following observations. Using the relevant confidence to set the threshold for OOD detection is novel and easy to implement.

(3) I appreciate the design of MixDiff and the detailed analysis. The empirical results demonstrate the proposed method could be useful in practice, especially when the knowledge of the model is largely restricted.

**Weaknesses:**

(1) The observation that OOD samples are less robust to perturbations (e.g., data augmentation methods such as MixUp) seems to align with the observation in the context of data poisoning, where poisoned data (can be also regarded as OOD samples) could be screened by strong augmentation methods. Although this is not strongly correlated, I encourage the authors to add a bit of discussion regarding this.

(2) Moreover, it would be interesting to see if MixDiff can be used as a defense against data poisoning. Of course, I am not asking the authors to perform relevant experiments, but it could be an interesting future direction.

**Questions:**

I don't have additional questions.

---

> ### Author Response · Authors · 2023-11-23
> **Response to Reviewer LGSq**
>
> Thank you for these insightful comments suggesting interesting connections to data poisoning attack. We are very glad that you found our approach well-motivated and novel. Below, we discuss the suggestions you have kindly provided to us in the review.
>
> > (1) The observation that OOD samples are less robust to perturbations  (e.g., data augmentation methods such as MixUp) seems to align with the  observation in the context of data poisoning, where poisoned data (can  be also regarded as OOD samples) could be screened by strong  augmentation methods. Although this is not strongly correlated, I encourage the authors to add a bit of discussion regarding this.
> >
>
> We appreciate the suggestion and have revised Section 2 to include such connections to data poisoning attack [(Section 2, page 3)]. Prior works [1, 2] suggest that data augmentation with Mixup reduces the attack efficacy by allowing the victim model to observe some parts of the poisoned samples’ adversarial features in none-poisoned samples during training, which is made possible by the interpolation of samples in the case of Mixup, or insertion of certain parts of an image in the case of CutMix. This may be viewed in another perspective, where poisoned samples’ adversarial features are perturbed by the Mixup operation. Attackers want the victim model to associate these artificial distractors with a certain outcome, i.e., being uncertain even when a sample exhibits the ground truth class characteristics, so that the class boundaries would be blurry. The two approaches seem to share certain characteristics, in the sense that the Mixup operation was used to reduce the adverse effect of distractors.
>
> > (2) Moreover, it would be interesting to see if MixDiff can be used  as a defense against data poisoning. Of course, I am not asking the  authors to perform relevant experiments, but it could be an interesting future direction.
> >
>
> It would certainly be an interesting direction to take, and see if MixDiff can be extended to such scenarios. A naive approach would be to apply a zero-shot CLIP classifier equipped with the given train set’s class names, and screening out certain train samples that are deemed to be abnormal. Inspired by your comment, we have performed an additional experiment [(Section 4.5, Table 4, pages 8-9), (Appendix E.6, page 21)] that is similar to this scenario, but in this case, an attacker is trying to fool the OOD detector, so that an ID sample is detected as OOD and vice versa. This is commonly done by adding a small modification to an image that is indistinguishable to human eye. The experimental results show that MixDiff is significantly more robust to such adversarial samples compared to other output-based scores, as can be seen in the AUROC scores in the table below. This suggests that our intuition that contributing features in misclassified samples are less robust to perturbations also holds for these artificially injected features that are designed to fool an OOD detector. Seeing as that data poisoning attacks share a certain degree of similarity to adversarial attacks in OOD detection, it would be interesting to explore an extension of our framework to such scenarios.
>
> | Method | CIFAR10 |  |  |  | CIFAR100 |  |  |  |
> | --- | --- | --- | --- | --- | --- | --- | --- | --- |
> |  | Clean | In | Out | Both | Clean | In | Out | Both |
> | Entropy | 89.88 | 47.42 | 13.77 | 2.678 | 79.87 | 36.86 | 14.38 | 2.21 |
> | MixDiff+Entropy | 90.64 | 54.71 | 31.77 | 9.084 | 81.11 | 47.42 | 31.40 | 9.08 |
> | MixDiff Only | 88.16 | 61.00 | 40.28 | 20.45 | 78.05 | 58.84 | 44.19 | 27.48 |
>
> [1] Dp-instahide: Provably defusing poisoning and backdoor attacks with differentially private data augmentations, 2021
>
> [2] Strong data augmentation sanitizes poisoning and backdoor attacks without an accuracy tradeoff. pp. 3855–3859, 06 2021a. doi: 10.1109/ICASSP39728.2021.9414862.

---

### Official Review · Reviewer_Z8Qr · 2023-11-01

**Soundness:** 2 fair
**Presentation:** 3 good
**Contribution:** 2 fair
**Rating:** 5
**Confidence:** 4

**Summary:**

- The authors address the issue of detecting out-of-distribution (OOD) data when there is no privileged access to model parameters or their activation. They begin by introducing an intuition that features contributing to misclassified samples (both ID and OOD) are more susceptible to perturbations. Building upon this insight, the authors propose a method called MixDiff, which involves mixing target and ID samples (oracle samples) with auxiliary samples to perturb both types of samples. By comparing the model outputs of the perturbed target samples with those of the oracle samples, the authors determine whether the target samples are OOD data. Experimental results across multiple datasets demonstrate the effectiveness of their method in achieving OOD detection while solely relying on the model output.

- In summary, leveraging the observation that contributing features in misclassified samples exhibit higher sensitivity to perturbations, the authors present MixDiff, a perturbation-based approach for OOD data detection. The method combines mixed samples and model output comparisons to effectively identify OOD data, as validated through experiments on various datasets.

**Strengths:**

- The authors explore a practical approach to OOD detection that relies solely on model inputs and outputs, offering significant value in real-world scenarios.
- They provide explicit theoretical and empirical evidence to support their method, showcasing its applicability through experiments involving out-of-scope (OOS) detection on an intent classification task.

**Weaknesses:**

- The existing experiments are not comprehensive enough. I would recommend the authors to include an additional experiment to address the following concern:
- While the experiments and theory conducted by the authors do support the effectiveness of their method, their motivation is based on the intuition that contributing features in misclassified samples are more sensitive to perturbations. However, this intuition lacks proper support and validation. Hence, it would be beneficial for the authors to incorporate relevant verification experiments to provide further evidence and clarify this aspect.

- The description of the algorithm section could be improved to enhance its intuitiveness. It would be helpful to include a schematic diagram illustrating the algorithm, providing readers with a clearer understanding of its workflow.
- In the related work section, the authors mention that some methods are not suitable for black-box APIs, where only access to inputs and outputs is available. An example of such a method mentioned in the paper is:
"Decoupling maxlogit for out-of-distribution detection." In Proceedings of the IEEE/CVF Conference on Computer Vision and Pattern Recognition (CVPR), pp. 3388–3397, June 2023.
- However, including a comparison with such methods would help readers understand the extent to which the proposed approach differs from ideal scenarios with more information, under the constraints of rigorous black-box conditions. This would provide further insights into the progress of OOD detection under black-box settings.

**Questions:**

I would like the author to supplement and polish the article based on the weaknesses.

---

> ### Author Response · Authors · 2023-11-23
> **Response to Reviewer Z8Qr [1/2]**
>
> We are grateful for such thoughtful and detailed feedback on our work. We find these comments to be really helpful. We are also very pleased that you found our approach offers a significant value in real-world scenarios where access of the model may not be always assumed. We would like to address the concerns you have raised in your review.
>
> > While the experiments and theory conducted by the authors do support the effectiveness of their method, their motivation is based on the intuition that contributing features in misclassified samples are more sensitive to perturbations. However, this intuition lacks proper support and validation. Hence, it would be beneficial for the authors to incorporate relevant verification experiments to provide further evidence and clarify this aspect.
> >
>
> Thank you for pointing out this important aspect. We have included a verification experiment in [(Section 1, Figure 1b, page 2), (Appendix E.8, Figure 6, pages 21-23)] to provide evidence for the intuition that contributing features in a misclassified sample are more sensitive to perturbations. In the experiment, we demonstrate that the locations in a sample on which the model has focused are more sensitive to perturbations when these features do not belong to the ground truth class. More specifically, for a given OOD sample with a high confidence score, we calculated the distance of the class activation maps of the predicted class before and after a perturbation. Compared to the class activation maps of the OOD sample’s actual ground truth class, these predicted class’s class activation maps tended to be more brittle to perturbations, despite having an initial high confidence prediction score for that class. This suggests that even though certain features in a misclassified OOD sample had strong impact on the model’s high degree of confidence, these features have tendency to be less robust to perturbations.
>
> We also have conducted another additional experiment that further clarifies this aspect [(Section 4.5, Table 4, pages 8-9), (Appendix E.6, page 21)]. The experimental setup is based on adversarial attack scenarios. In an adversarial attack on an OOD detector, attackers aim to fool the OOD detector, so that an ID sample is detected as OOD, and vice versa. This is commonly done by adding a small modification to a target sample that is indistinguishable to human eye. It can be viewed that these modifications inject certain artificial features in the target sample so as to induce overconfident or underconfident outputs from the model. The experimental results demonstrate  MixDiff’s superiority on robustness to adversarial attacks. This indicates that the contributing features that induce ID/OOD misclassification are less robust to perturbations and that MixDiff can effectively exploit such brittleness. We present the results below.
>
> | Method | CIFAR10 |  |  |  | CIFAR100 |  |  |  |
> | --- | --- | --- | --- | --- | --- | --- | --- | --- |
> |  | Clean | In | Out | Both | Clean | In | Out | Both |
> | Entropy | 89.88 | 47.42 | 13.77 | 2.678 | 79.87 | 36.86 | 14.38 | 2.21 |
> | MixDiff+Entropy | 90.64 | 54.71 | 31.77 | 9.084 | 81.11 | 47.42 | 31.40 | 9.08 |
> | MixDiff Only | 88.16 | 61.00 | 40.28 | 20.45 | 78.05 | 58.84 | 44.19 | 27.48 |
>
> We also have revised Figure 1 in [(Section 1, Figure 1a, page 2)] to better illustrate our motivation by displaying the change in the class activation maps of a high-confidence OOD sample while it undergoes a perturbation, and its corresponding oracle (an ID sample with the same predicted class) sample’s class activation maps under the same perturbation.
>
> > The description of the algorithm section could be improved to enhance its intuitiveness. It would be helpful to include a schematic diagram illustrating the algorithm, providing readers with a clearer understanding of its workflow.
> >
>
> Thank you for your interest in our research and for your detailed review. We are particularly grateful for your feedback on enhancing the explanation of our algorithm, which is a crucial component of understanding the structure of our methodology.
>
> We have revised Figure 2 in the manuscript to provide a more intuitive and integrated understanding of the flow of the algorithm [(Section 3, Figure 2, page 4)]. Whereas the original Figure 2 was intended to provide an intuitive view of the overall flow of how the MixDiff score is computed, this revised Figure 2 reflects the order in which the Algorithm 1 is followed, showing how MixDiff works step-by-step. By presenting a simple example in detailed steps, Figure 2 has become more intuitive and easier to understand the proposed method.

---

> ### Author Response · Authors · 2023-11-23
> **Response to Reviewer Z8Qr [2/2]**
>
> > In the related work section, the authors mention that some methods are not suitable for black-box APIs, where only access to inputs and outputs is available. An example of such a method mentioned in the paper is: "Decoupling maxlogit for out-of-distribution detection." In Proceedings of the IEEE/CVF Conference on Computer Vision and Pattern Recognition (CVPR), pp. 3388–3397, June 2023.
> However, including a comparison with such methods would help readers understand the extent to which the proposed approach differs from ideal scenarios with more information, under the constraints of rigorous black-box conditions. This would provide further insights into the progress of OOD detection under black-box settings.
> >
>
> Thank you for this valuable suggestion. This would indeed help the readers to assess the state of OOD detection task in black-box environments. To that end, we have run additional experiments  [(Section 4, Table 1, page 8)] with two such methods which rely on model’s inner activations, namely  DML [1] and ASH [2] and added the results in Table 3. For fair comparison with zero-shot OOD detection methods, we applied these methods to zero-shot CLIP classifier. However, these methods did not outperform the baseline scores. We suspect this is because they were validated with finetuned models such as ResNet50 finetuned on the ID dataset, whereas our experimental setup focuses on zero-shot OOD detection. This result is in line with previous findings [3] where a similar method, ReAct [4], did not outperform the Energy score in zero-shot OOD detection task.
>
> This has prompted us to run another experiment [(Section 4, Table 1, page 7)] to evaluate MCM [5]. MCM has shown strong performance on zero-shot OOD detection task. However, it requires an access to the logit space, which may not be accessible in a black-box API. This method outperforms the baseline scores that are based on the model’s prediction probabilities, namely, Entropy and MSP, as well as two other logit-based OOD scores (MLS, Energy). However, MixDiff equipped with entropy score outperforms MCM, on 4 out of 5 datasets as well as on average over the 5 datasets, while solely relying on model’s prediction probabilities. The results suggest that MixDiff’s perturb-and-compare approach is effective at gathering more discriminative information that are not available in prediction probabilities alone. Moreover, MixDiff further improves MCM’s detection performance when applied to it. We present the average AUROC scores of each method in the table below.
>
> | Method | CIFAR10 | CIFAR100 | CIFAR+10 | CIFAR+50 | TinyImageNet | Average |
> | --- | --- | --- | --- | --- | --- | --- |
> | MCM (Ming et al., 2022) | 90.6±2.9 | 80.3±2.1 | 96.9±0.8 | 97.0 | 83.1±2.2 | 89.6 |
> | MixDiff+MCM | 91.4±1.8 | 81.4±2.6 | 97.5±0.9 | 97.7 | 83.9±2.2 | 90.4 |
> | Entropy | 89.9±2.6 | 79.9±2.5 | 96.8±0.8 | 96.8 | 82.2±2.3 | 89.1 |
> | MixDiff+Entropy (Pred. prob.) | 91.1±1.6 | 80.9±2.6 |  97.1±0.8 | 97.3 | 82.9±2.3 | 89.9 |
> | DML (Zhang & Xiang, 2023) | 87.8±3.0 | 80.0±3.1 | 96.1±0.8 | 96.0 | 84.0±1.2 | 88.8 |
> | ASH (Djurisic et al., 2023) | 85.2±3.8 | 75.4±4.4 | 92.5±0.9 | 92.4 | 77.2±3.1 | 84.5 |
>
> [1] Decoupling maxlogit for out-of-distribution detection. In Proceedings of the IEEE/CVF Conference on Computer Vision and Pattern Recognition (CVPR), pp. 3388–3397, June 2023.
>
> [2] Extremely simple activation shaping for out-of-distribution detection. In The Eleventh International Conference on Learning Representations, 2023.
>
> [3] Clipn for zero-shot ood detection: Teaching clip to say no. In Proceedings of the IEEE/CVF International Conference on Computer Vision, pp. 1802–1812, 2023.
>
> [4] React: Out-of-distribution detection with rectified activations. In Advances in Neural Information Processing Systems, 2021.
>
> [5] Delving into out-of-distribution detection with vision-language representations. Advances in Neural Information Processing Systems, 35:35087–35102, 2022.

---

### Author Response · Authors · 2023-11-23
**Summary of additional experiments and updates**

We would like to thank the reviewers for their constructive feedback and inspiring comments. With these comments, we were able to strengthen and more clearly highlight our work’s overall contribution. Below are the list of additional experiments and revised contents that we would like to share with the reviewers.

- **Experiments on robustness to adversarial attacks [(Section 4.5, Table 4, pages 8-9), (Appendix E.6, page 21)]:** We have added results of adversarial attack in OOD detection experiments, thanks to the insightful comments from *Reviewers LGSq, hNhc*.
- **Verification experiment of the main motivation [(Section 1, Figure 1b, page 2), (Appendix E.8, Figure 6, pages 21-23)]:** We added justification for our intuition that contributing features that induce misclassification are less robust to perturbations, thanks to *Reviewers Z8Qr, hNhc, EmiV.*
- **Illustration of motivation figure [(Section 1, Figure 1a, page 2)]:** We have modified the figure to better illustrates the main motivation by highlighting the focused regions with class activation map, thanks to the comments from *Reviewers Z8Qr, hNhc, EmiV.
- **Computational cost analysis [(Appendix E.11, Figure 10, pages 24-25)]:** We demonstrate that MixDiff’s computational cost can be adjusted to fit a given computational budget while still enjoying the performance increase from a base score, incorporating the suggestions from *Reviewers hNhc,  EmiV***.**
- **Processing time analysis [(Appendix E.12, Figure 11, pages 24-25)]:** We show that MixDiff framework allows a high degree of parallelism when computing the additional perturbed samples, this reflects the feedbacks from *Reviewers hNhc,  EmiV*.
- **Refined approach in theoretical analysis [(Section 3.1, Theorem 1, pages 5-6), (Appendix C, Theorem 1, pages 17-18)]:** We have refined our approach to proof of Theorem 1, and now take more relaxed assumptions, thanks to the feedback from *Reviewer hNhc*.
- **Addition of other OOD detection baseline methods [(Section 4, Table 1, pages 6-8)]:** We have included more recent and strong baseline methods for OOD detection, including methods that assume a white-box classifier model, incorporating suggestions from *Reviewers Z8Qr, hNhc.*
- **Revision of the main figure [(Section 3, Figure 2, page 4)]:** We revised the main figure to illustrate the steps involved when computing MixDiff score, this improvement reflects the feedback from *Reviewer hNhc.*
- **Demonstration of MixDiff’s contribution to other OOD scores [(Appendix E.9, Figure 8, pages 23-24)]:** We included a qualitative analysis that illustrates that MixDiff scores can help detect a pair of ID and OOD samples when the model’s outputs are almost identical, thanks to *Reviewer EmiV*.
- **MixDiff in relation to adversarial attacks [(Section 2, page 3)]:** We were able to more clearly position MixDiff among other related areas such as data poisoning and adversarial attack on OOD detectors, with the insightful feedback form *Reviewers LGSq, hNhc.*

We believe these results would not have been possible without the reviews and are grateful for the feedback that we have received. We regret that we could not finish these additional experiments early, missing the valuable opportunity of further discussion with the reviewers.

---

### Meta-Review · Area_Chair_puMs · 2023-12-02

**Metareview:**

This paper introduces an out-of-distribution detection approach in a black-box setting, where a model is accessible only via API calls so that the user has no access to parameters or gradients, but gets only class log probabilities as an output once the model is queried with a data instance. The proposal is built upon the intuition that input perturbations affect in- and out-distribution data differently so, provided that one has access to a known-to-be-in-distribution sample, it's then possible to compare the effects on the outputs of perturbed types of data to decide when not to trust a prediction.

While the reviewers recognized the value in the proposal of detection methods that rely only on limited information, the paper has limitations that prevent its acceptance in its current form, and we highlight some of those limitations in the following. The presentation needs additional information since the idea of an out-distribution is ill-defined in general, and it's up to the authors to properly set a scope and define what exact distribution shifts are being considered. Moreover, the notation is a bit confusing and not well defined. In terms of the methodology, it's worth highlighting issues such as the fact that upon computing the proposed scores, the proposed method then adds them to some other existing detection score after multiplying by a hyperparameter, so the intuition upon which the idea is built is somewhat questionable since the perturbation effect has no discriminative effect on its own. The effect of the selection of oracle samples is a bit unclear too, and so is the impact of that choice. The reason for the choice of mixup as a perturbation is also unclear, and what authors refer to as mixup in the case of text, is in fact something else. I would finally highlight that the proposal incurs significant overhead requiring multiple forward passes for score computation.

**Justification For Why Not Higher Score:**

While the paper is quite interesting, there are issues that should be addressed before it being ready for publication as outlined in the latter portion of the meta-review.

**Justification For Why Not Lower Score:**

N/A

---

### Decision · Program_Chairs · 2024-01-16

Reject